# Melatonin Prevents Chondrocyte Matrix Degradation in Rats with Experimentally Induced Osteoarthritis by Inhibiting Nuclear Factor-κB via SIRT1

**DOI:** 10.3390/nu14193966

**Published:** 2022-09-24

**Authors:** Mingchao Zhao, Xiaopeng Song, Hong Chen, Tianwen Ma, Jilang Tang, Xinyu Wang, Yue Yu, Liangyu Lv, Lina Jia, Li Gao

**Affiliations:** Heilongjiang Key Laboratory Animals and Comparative Medicine, College of Veterinary Medicine, Northeast Agricultural University, Harbin 150030, China

**Keywords:** melatonin, chondrocytes, SIRT1, NF-κB pathway, TGF-β1/Smad2 pathway

## Abstract

Osteoarthritis (OA) is a common degenerative joint disease characterized by an imbalance of cartilage extracellular matrix (ECM) breakdown and anabolism. Melatonin (MT) is one of the hormones secreted by the pineal gland of the brain and has anti-inflammatory, antioxidant, and anti-aging functions. To explore the role of MT in rats, we established an OA model in rats by anterior cruciate ligament transection (ACLT). Safranin O-fast green staining showed that intraperitoneal injection of MT (30 mg/kg) could alleviate the degeneration of articular cartilage in ACLT rats. Immunohistochemical (IHC) analysis found that MT could up-regulate the expression levels of collagen type II and Aggrecan and inhibit the expression levels of matrix metalloproteinase-3 (MMP-3), matrix metalloproteinase-13 (MMP-13), and ADAM metallopeptidase with thrombospondin type 1 motif 4 (ADAMTS-4) in ACLT rats. To elucidate the mechanism of MT in protecting the ECM in inflammatory factor-induced rat chondrocytes, we conducted in vitro experiments by co-culturing MT with a culture medium. Western blot (WB) showed that MT could promote the expression levels of transforming growth factor-beta 1 (TGF-β1)/SMAD family member 2 (Smad2) and sirtuin 2-related enzyme 1 (SIRT1) and inhibit the expression of levels of phosphorylated nuclear factor of kappa light polypeptide gene enhancer in B-cells inhibi-tor (p-p65) and phosphorylated IκB kinase-α (p-IκBα). In addition, WB and real-time PCR (qRT-PCR) results showed that MT could inhibit the expression levels of MMP-3, MMP-13, ADAMTS-4, inducible nitric oxide synthase (iNOS), and cyclooxygenase-2 (COX-2) in chondrocytes induced by interleukin-1β (IL-1β), and up-regulate the expression of chondroprotective protein type II collagen. We found that in vivo, MT treatment protected articular cartilage in the rat ACLT model. In IL-1β-induced rat chondrocytes, MT could reduce chondrocyte matrix degradation by up-regulating nuclear factor-kB (NF-κB) signaling pathway-dependent expression of SIRT1 and protecting chondrocyte by activating the TGF-β1/Smad2 pathway.

## 1. Introduction

Osteoarthritis (OA) is a degenerative disease caused by genetic susceptibility, mechanical damage, aging, and other reasons [1]. OA is a major skeletal disease leading to disability after middle age, which affects the quality of life of patients [2,3]. OA is not easily diagnosed in the early stage. The main clinical symptoms are difficulty in movement caused by pain and cartilage degeneration [4]. At present, the clinical treatment of OA centers mainly on pain relief, and there is no effective method to inhibit the process of OA [5]. Joint replacement is the only way to treat advanced OA, but artificial joints still have problems, such as high cost and equipment aging. Therefore, it is of great significance to promote cartilage repair and delay the progression of OA.

Melatonin (MT) is one of the hormones secreted by the pineal gland. It is a broad-spectrum antioxidant and free radical scavenger [6,7,8], which can achieve antioxidant effects by scavenging reactive oxygen species (ROS) and reactive nitrogen species (RNS) [9]. It is widely used in various fields due to its powerful antioxidant, anti-inflammatory, and anti-aging properties [10,11,12]. In animal models, such as those of rats and pigs, and in in vitro experiments, efficacy of melatonin has been demonstrated following both intra-articular and intra-peritoneal injections. In these models, melatonin can maintain the activity of chondrocytes, promote the synthesis of articular chondrocyte matrix, and protect articular cartilage and chondrocytes [13,14,15,16]. MT may exert protective effects through multiple pathways. However, the exact role of MT in the progression of OA, especially the regulatory mechanisms in chondrocytes, remains unclear. Sirtuin 2-related enzyme 1 (SIRT1) is a nicotinamide adenine dinucleotide (NAD)-dependent nuclear histone deacetylase involved in the regulation of senescence, apoptosis, inflammation, and deacetylation. The deacetylation ability of SIRT1 can regulate a variety of pathways, such as regulating nuclear factor-kB (NF-kB), AMP-activated protein kinase (AMPK), and other pathways, to inhibit the expression of cartilage degrading enzymes and play a protective role [17,18]. SIRT1 is involved in the regulation of chondrocyte gene expression and plays an important role in cartilage repair, and delaying the progression of OA, and may be an important target to control the progression of OA. Experiments show that MT exerted protective and anti-inflammatory effects on hydrogen peroxide (H_2_O_2_)-stimulated human chondrocytes and rabbit OA model chondrocytes by activating SIRT1 [19]. However, another experiment found that MT in human chondrocytes could alleviate OA by inhibiting the enhancement of SIRT1 [20]. It can be seen that previous studies have found that there is an interactive relationship between MT and SIRT1, but there are opposite experimental results. Therefore, this experiment is expected to explore the mechanism of MT and SIRT1 in IL-1β-induced chondrocytes. NF-κB is an important intracellular nuclear transcription factor that plays a key role in regulating cell survival genes and coordinating cytokine expression [21]. In OA, NF-κB can activate the expression of destructive mediators, such as matrix metalloproteinases (MMPs), ADAM metallopeptidase with thrombospondin type 1 motif 4 (ADAMTS-4), and pro-inflammatory factors, including inducible nitric oxide synthase (iNOS), cyclooxygenase-2 (COX-2), and prostaglandin E2 (PGE2) [22,23]. MT can inhibit the production of tumor necrosis factor-alpha (TNF-α) and interleukin 1-beta (IL-1β), by downregulating the NF-κB signaling pathway, and the overexpression of a microRNA, miR-3150a-3p, thereby reducing the inflammatory response [24,25,26]. In chondrocytes, MT can inhibit the nuclear translocation of NF-κB p65 in H2O2-stimulated chondrocytes and inhibit the expression of ROS and MMPs in synovial mesenchymal stem cells, thereby exhibiting anti-inflammatory effects and promoting cartilage formation [27]. Our study aimed to explore the interaction between the MT/SIRT1/NF-κB signaling pathway in IL-1β-induced chondrocytes. Transforming growth factor-β (TGF-β) can promote tissue repair and regeneration and plays an important role in cartilage formation and articular cartilage maintenance [28,29,30]. The high expression of transforming growth factor-beta 1 (TGF-β1) can stimulate the transcription of collagen and proteoglycan core proteins, inhibit the transcription level of MMPs, and induce the synthesis of metalloproteinase inhibitors, stimulating the proliferation of chondrocytes and achieving a protective effect on cartilage [31]. A previous study showed that intra-articular injection of MT solution could inhibit the development of OA, which might be related to promoting the expression of TGF-β1 in cartilage [32]. However, the mechanism of the TGF-β pathway in OA remains unclear.

The purpose of the underlying study was to investigate the protective effect and mechanism of MT on chondrocytes. Based on previously reviewed data, we chose to study the SIRT1/NF-κB/TGF-β1/Smad family member 2 (Smad2) pathway. We sought to elucidate the mechanism of action of MT in chondrocytes to provide new therapeutic options for reducing the degradation of the chondrocyte matrix and slowing the progression of OA.

## 2. Materials and Methods

### 2.1. Reagents and Antibodies

MT, Rat IL-1β, 4′,6-diamidino-2-phenylindole (DAPI), collagenase type II, and EX527 were purchased from Sigma Aldrich (St. Louis, MO, USA). Fetal bovine serum (FBS) (BI, Kibbutz Bet Hamek, Israel), Dulbecco’s Modified Eagle’s Medium-(DMEM-) F12 Medium, and Phosphate Buffered Saline purchased from Corning (New York, NY, USA). 1% penicillin/streptomycin, cell counting kit-8 (CCK-8), and staining kit were purchased from Beyotime (Haimen, China). Trypsin (0.25%) was purchased from Gibco (Grand Island, NY, USA). Lysate and RAPI were purchased from Biyuntian (Shanghai, China). nuclear factor of kappa light polypeptide gene enhancer in B-cells inhibitor (p65), Phosphorylated nuclear factor of kappa light polypeptide gene enhancer in B-cells inhibitor (p-p65),alpha (IκBα), p-IκBα, TGF-β1, Smad2, iNOS, COX-2, matrix metalloproteinase-3 (MMP-3), matrix metalloproteinase-13 (MMP-13), collagen type II alpha 1 chain (COL2A1), SIRT1, and ADAMTS-4 were purchased from ABclonal Technology (Wuhan, China). Horseradish peroxidase-conjugated goat anti-mouse/rabbit immunoglobulin G (IgG) secondary antibody was purchased from ABclonal Technology (UK). GAPDH was purchased from CST (Danvers, MA, USA).

### 2.2. Animal Ethics

Sprague-Dawley (SD) rats were purchased from the Animal experiment base of Harbin Medical University, Harbin, China. Rats used for cell experiments were 2–3 weeks old (self-bred). Adult males (*n* = 30, 300 ± 20 g) were used in the surgical model. The animal care and use involved in this study were approved by the protocol of the Animal Welfare Committee of Northeast Agricultural University (Harbin, China). All efforts were aimed at minimizing animal suffering and reducing the number of animals used (Number: #NEAU-2020-07-1709-12).

### 2.3. Surgical Model of Anterior Cruciate Ligament Transection (ACLT)

Experimental rats were operated on according to the modeling method for OA animals [33]. Thirty 7-week-old male SD (weight of 300–350 g) rats were purchased from Harbin Medical University. The rats were randomly divided into 3 groups: sham, model, andMT (30 mg/kg) groups. In the sham group, only the skin was incised to open the joint cavity, but the cruciate ligament was not damaged. ACLT was performed on the right knee joint of 20 SD rats as a model group and MT (30 mg/kg) group [15], respectively. After the operation, the rats were kept in separate cages, and the room was kept clean, at constant temperature (23 ± 1 °C), and ventilated. Animals were given adequate water and food during the rearing period, and the bedding was changed every 4 d. After 24 h had elapsed since the operation, the MT group was administered MT by intraperitoneal injection once every two days for 12 consecutive weeks. After dissolving MT with minimal dimethyl sulfoxide (DMSO), it was diluted to 30 mg/kg with saline. The sham group and the model group were intraperitoneally injected with a mixture of DMSO and saline. After 12 weeks, experimental materials, such as the right knee joint and serum, were collected.

### 2.4. Histological and Immunohistochemical (IHC) Analyses

After 12 weeks of modeling, samples of the right hind limb of each group of rats were taken and stained with Safranin O-fast green to observe the pathological changes in articular cartilage. This was scored using an adapted version of the Mankin score [34]. The Mankin score was based on four aspects: the integrity of cartilage structure, the distribution of chondrocytes, the staining of the cartilage matrix, and the integrity of the tide line. The higher the score, the more severe the cartilage degeneration. In IHC analysis, rabbit primary antibody and a diaminobenzo (DAB) substrate kit were used to reflect the situation of each indicator semi-quantitatively, and buffer was used as a negative control instead of primary antibody. Under the pathological microscope, the expression levels of MMP-3, Aggrecan, MMP-13, ADAMTS-4, and type II collagen appeared yellow-brown in the cartilage extracellular matrix (ECM) of chondrocytes. We analyzed data from all groups using Image-Pro Plus software version 6.0 (Media Cybernetics, Rockville, MD, USA).

### 2.5. Enzyme-Linked Immunosorbent Assay (ELISA) of Serum

Rat serum was obtained after centrifugation (2500 rpm/min, 20 min), and the rat serum MT, iNOS, COX-2, PGE2, and TNF-α assay kits (NJJCBIO, Nanjing, China) were used for detection according to the manufacturer’s instructions.

### 2.6. Isolation and Culture of Chondrocytes

All SD rats were kept under specific pathogen-free conditions to extract primary chondrocytes from the bilateral femoral head and tibial plateau cartilage of rats that were 14-21 days old. The rat articular cartilage was cut into slices, digested with 0.25% trypsin for 30 min, and digested with 0.2% collagenase type II in DMEM-F12 for 4 h at 37 °C. The chondrocyte suspension was centrifuged at 1200 rpm for 5 min. After centrifugation, chondrocytes were resuspended in DMEM-F12 complete medium supplemented with 10% FBS and 1% penicillin and streptomycin. The resuspended chondrocytes were aliquoted into cell culture flasks (Corning, Corning, NY, USA) and cultured at 37 °C in a 5% CO_2_ incubator (Thermo Fisher, Waltham, MA, USA). To minimize phenotypic loss, in vitro experiments were performed using rat chondrocytes no later than the second passage.

### 2.7. Cell Proliferation Assay

Cell proliferation assay was assessed by detecting dehydrogenase in viable cells using 96-well plates (5 × 10^3^ cells/well). The chondrocytes containing only DMEM-F12 and 10% FBS were used as the control group, and the chondrocytes induced with 10 ng/mL IL-1β for 24 h were used as the model group. The experimental group was induced with 10 ng/mL IL-1β and different concentrations of MT (100, 200, 400, 800, 1600 ng/mL) for 24 h and 48 h. MT was dissolved in DMSO and then diluted to the target concentration with a culture medium (containing 10% FBS and 1% antibiotics). Since the concentration of DMSO was much less than 0.1%, it was safe for cells. Thus, DMSO was not used as a control group. After 4 h of co-incubation at 37 °C in the dark, the absorbance was measured at 450 nm using a microplate reader.

### 2.8. Western Blot (WB)

Whole-cell proteins were extracted using lysis buffer (Beyotime, Shanghai, China) containing phosphatase inhibitors (Beyotime, Shanghai, China). Chondrocytes containing DMEM-F12 and 10% FBS were in the control group and were induced with 10 ng/mL IL-1β for 24 h as the model group. The MT group was induced with 10 ng/mL IL-1β and different concentrations of MT (100, 200, 400, 800 ng/mL) for 24 h. The inhibitor group was treated with 10 ng/mL IL-1β, MT (800 ng/mL), and SIRT1 inhibitor EX527 (20 uM) for 24 h [35]. Half an hour after adding the lysate, it was transferred to a centrifuge tube and centrifuged at 12,000 rpm for 15 min at 4 °C, and the supernatants of each group were collected. The protein concentration of chondrocytes in each group was determined by a BCA protein detection kit (Thermo Fisher, Waltham, MA, USA). Total proteins of different groups of cells were separated by electrophoresis and transferred to polyvinylidene fluoride (PVDF) filters. Then, the cells were blocked with 5% bovine serum albumin (BSA)/skimmed milk powder (Wandashan, Hebei, China) for 1.5 h at room temperature and incubated with primary antibody for 12 h at 4 °C: p-65, p-p65, IκBα, TGF-β1, Smad2, iNOS, COX-E, MMP-3, MMP-13 (1:1000, Novus, Littleton, CO, USA), ADAMTS-4 (1:500, ABCAM, Cambridge, UK), COL2A1, SIRT1, and GAPDH (1:1000, CST, Danvers, MA, USA). PVDF was then washed three times with tris buffered saline with tween 20 (TBST) and incubated with horseradish peroxidase-conjugated goat anti-mouse/rabbit IgG secondary antibody for 60 min at room temperature. The membrane was washed 3 times with TBST and detected using the ECL system, and the gray value was quantified with Image (1.8.0, National Institute of Mental Health, AK, USA, https://imagej.en.softonic.com/) J software.

### 2.9. Real-Time PCR (qRT-PCR) 

Total RNA was extracted using an RNA extraction kit (Tiangen, Beijing, China), according to the manufacturer’s instructions. Complementary DNA (cDNA) was synthesized by reverse transcription using Prime Script cDNA RT Master Mix (Kogen, Tokyo, Japan). Amplification was performed with RNase DDH_2_O (Tiangen, Beijing, China), SYBR PreMix Ex Taq™ (Kogen, Tokyo, Japan), and forward and reverse primers for the gene of interest. The entire PCR process was monitored using a LightCycler^®^ 480 (Roche, Germany), and cycle threshold (CT) relative expression levels were calculated using the 2^−ΔΔCT^ method. The primer sequences are shown in Table 1. Primer sequence.

### 2.10. Immunofluorescence Staining (IF)

Each dish was fixed with 1 mL of 4% paraformaldehyde (Solarbio, Beijing, China) for 25 min, permeabilized with 0.2% Triton X-100 (Solarbio, Beijing, China) for 0.5 h, and blocked with 5% BSA for 1.5 h at room temperature. The primary antibodies of p-p65, p-Smad2, COL2A1, and SIRT1 were added to the plate and incubated at 4 °C for 24 h. Then, the Alexa Fluor488-conjugated secondary antibody was added to PBS at a dilution of 1:500 and incubated in the dark for 1.5 h. Finally, the DAPI solution was added at room temperature and incubated in the dark for 10 min. Positively stained cells in each image were detected and analyzed using an inverted fluorescence microscope (Nikon, Tokyo, Japan).

### 2.11. Statistical Analysis

In vitro experiments were performed at least 3 times. Our results are expressed as mean ± standard deviation (SD). We used Bartlett’s test to examine variance between samples (*p* > 0.05), followed by one-way analysis of variance (ANOVA) with Dunnett’s multiple comparisons test for multiple comparisons. *p* < 0.05 was considered statistically significant.

## 3. Results

### 3.1. MT Attenuates Inflammation and Chondrocyte ECM Degradation in OA Rats

The results of ELISA showed (Figure 1) that compared with the sham group, the model group significantly increased circulating levels of COX-2, iNOS, TNF-α, and PGE2. Compared with the model group, the MT group significantly decreased circulating levels of COX-2, iNOS, TNF-α, and PGE2.

The results of Safranin O-fast green staining showed (Figure 2A) that the surface of the articular cartilage in the sham group was smooth, without dents or damage, and the edges were flat. The lines were complete and clear. The chondrocytes in the model group were distributed in chondrons, and the red staining was significantly reduced. The number of hypertrophied cells increased significantly, and the tide lines were distorted and irregular. Compared with the model group, the hypertrophied cells in the MT group were significantly reduced. Although the chondrocytes were clustered, the matrix was uniformly stained red. The Mankin score of articular cartilage showed that the model group was significantly higher than the sham group, and the MT group was significantly lower than the model group. In IHC, we used buffer instead of primary antibody as a negative control to exclude the effect of endogenous substances and specific staining on the experimental results (Appendix A). IHC staining showed (Figure 2B) that collagen type II and Aggrecan were strongly expressed in the articular cartilage of the sham and MT groups, and the degree of brown staining was higher than that of the model group. On the contrary, MMP3, MMP13, and ADAMTS-4 were weakly positive in the articular cartilage of the sham and MT groups, and the degree of brown staining was lower than that of the model group. Meanwhile, the Mankin score agreed with the histological analysis result (Figure 2A).

### 3.2. Co-Culture of MT with IL-1β Promotes Chondrocyte Proliferation, and MT Promotes SIRT1 Expression in IL-1β-Induced Chondrocytes

CCK8 results showed no cytotoxic effect of MT co-culture with 10 ng/mL IL-1β on chondrocytes cultured in vitro (Figure 3A). OA is a chronic degenerative disease under long-term oxidative stress. Thus, we selected 10 ng/mL IL-1β to stimulate rat chondrocytes for 24 h in vitro. Therefore, in subsequent experiments, 100, 200, 400, and 800 ng/mL of MT and 10 ng/mL of IL-1β were selected for co-action for 24 h. To explore the effect of MT on SIRT1, we detected the expression level of SIRT1 protein by WB. As shown in Figure 3C,D, IL-1β decreased SIRT1 expression in rat chondrocytes, whereas MT treatment abolished the effect of IL-1β. The qRT-PCR results were consistent with WB results (Figure 3B).

### 3.3. MT Inhibits NF-κB Signaling and Activates TGF-β1/Smad2 Pathway in IL-1β-Induced Chondrocytes

To explore the mechanism of MT reducing the degradation of the chondrocyte matrix, we examined the effect of MT on the NF-κB signaling and TGF-β1/Smad2 pathways. The rat chondrocytes were induced with 10 ng/mL IL-1β and co-cultured with or without MT at various concentrations for 24 h. WB results (Figure 4A,B) showed that, compared with the IL-1β group, MT down-regulated the expression levels of p-p65 and p-IκBα proteins, but up-regulated the expression levels of TGF-β1 and Smad2 proteins. 

### 3.4. MT Inhibits the Expression of Matrix Degradation-Related Indicators in IL-1β-Induced Chondrocytes

We investigated whether MT could protect against ECM degradation in IL-1β-treated chondrocytes. Compared with the IL-1β group, WB (Figure 5A,C) and qRT-PCR (Figure 5B,D) showed that the levels of MMP-3, MMP-13, ADAMTS-4, iNOS, and COX-2 were significantly higher, and the level of collagen II was significantly increased in the MT-treated group. 

### 3.5. EX527 Abrogates MT Activation of SIRT1 in IL-1β-Induced Chondrocytes

To explore the role of MT on SIRT1, we used EX527, a known inhibitor of SIRT1. We detected the expression of SIRT1 protein in each group by WB. As shown in Figure 6A, IL-1β reduced SIRT1 expression in rat chondrocytes, and MT treatment abolished the effect of IL-1β. However, the addition of EX527 reversed this effect. The results of IF staining showed that the green brightness of the MT group was significantly increased, suggesting that the expression of SIRT1 protein was up-regulated. The addition of EX527 significantly reduced brightness, suggesting that EX527 abolished the activation of SIRT1 by MT. The results of qRT-PCR (Figure 6B,C) IF staining were consistent with WB results.

### 3.6. The Inhibitor EX527 Reverses the Inhibitory Effect of MT on the NF-κB Pathway but Does Not Affect the Activation of the TGF-β1/Smad2 Pathway

As shown in Figure 7A, both IL-1β and EX527 increased the expression levels of p-p65 and p-IκBα and decreased the expression of IκBα in rat chondrocytes. The results showed that MT inhibited upregulation of p-p65 and p-IκBα protein expression levels in IL-1β-induced chondrocytes, but EX527 reversed this effect, which demonstrated that MT might decrease the production of p-p65 and p-IκBα by inhibiting the degradation of IκBα. The results of IF staining (Figure 7C) were consistent with the results of WB (Figure 7A). MT significantly reduced the intensity of green fluorescence, suggesting that MT inhibited IL-1β-induced p-p65 entry into the nucleus, but the addition of EX527 changed this result. In addition, WB results showed (Figure 7B) that IL-1β inhibited the expression levels of TGF-β1 and Smad2, and the addition of MT changed this trend. However, adding EX527 did not change the activation of TGF-β1 and Smad2 by MT. The results of IF staining (Figure 7D) were the same as the results of WB (Figure 7B), and MT significantly enhanced the intensity of green fluorescence, suggesting that MT promoted the translocation of Smad2 into the nucleus, but EX527 did not have this effect.

### 3.7. EX527 Reverses the Inhibitory Effect of MT on IL-1β-Induced Chondrocyte Matrix Degradation-Related Markers

The results of WB and qRT-PCR (Figure 8A–D) showed that the expression levels of MMP-3, MMP-13, ADAMTS-4, iNOS, and COX-2 were increased in IL-1β-stimulated rat chondrocytes, and the expression of type II collagen was reduced. Compared with the IL-1β group, the IL-1β+ EX527 group showed the same trend. MT treatment obviously decreased the expression levels of MMP-3, MMP-13, ADAMTS-4, iNOS, and COX-2 and increased the expression level of type II collagen, and the addition of EX527 significantly reversed this trend. IF results showed that the green fluorescence intensity of the MT group was remarkably enhanced, suggesting that the content of COL2A1 was up-regulated, and the addition of EX527 reversed this trend. The results of type II collagen IF (Figure 8E) were consistent with those of WB and qRT-PCR. 

## 4. Discussion

Melatonin is an important hormone involved in the regulation of circadian rhythms, and several studies have shown that a decrease in systemic levels of melatonin can lead to circadian rhythm disorders [36,37]. In addition, circadian rhythm disorders are one of the causes of osteoarthritis, depression and many other diseases [38,39,40]. Here, we investigated the protective effect of melatonin on cartilage. To explore whether MT can effectively delay the progression of arthritis in vivo, we performed an intraperitoneal injection at a concentration of 30 mg/kg to test the effect of MT in the ACLT-induced experimental OA model. By ELISA, we found that the serum MT level of the rats in the MT group was increased. Compared with the model group, the levels of iNOS, COX-2, PGE2, and TNF-α were significantly decreased. The enzyme responsible for nitric oxide (NO) production is iNOS, which is associated with significantly increased NO levels. NO is an important inflammatory mediator that induces release, including that of MMP-3 and MMP-13, which are closely associated with the pathological development of OA [41,42,43]. In addition, TNF-α is a promoter of NO production and correlates with apoptosis and patient pain levels [44,45]. Previous studies have shown that MT and its metabolites may inhibit the production of NO, IL-1β, and TNF-α in the blood and reduce cartilage degradation through antioxidant and anti-inflammatory effects [46,47]. COX-2 is an inflammatory factor closely related to inflammation and pain [48]. It can be induced by factors, such as IL-1 and TNF-α, to show abnormally high expression [49,50]. PGE2, a metabolite of arachidonic acid cyclooxygenase, is an important pro-inflammatory mediator. At present, the main drug widely used in the clinical treatment of OA, non-steroidal anti-inflammatory drugs (NSAIDs), reduce the synthesis of PGE2 by inhibiting COX-2, thereby achieving anti-inflammatory and analgesic effects. Therefore, we suggest that MT may have anti-inflammatory, analgesic, and anti-cartilage destruction effects in a rat model of OA.

The results of Safranin O fast green staining showed that, compared with the model group, the morphology and extracellular matrix of chondrocytes in the MT group were relatively intact. MT can protect articular cartilage in ACLT-induced osteoarthritis rats. Zhang Yi et al. reported that intra-articular injection of MT successfully slowed the progression of surgically-induced OA by regulating the homeostasis of the cartilage matrix [14]. Studies have found that MT can increase the expression levels of cartilage markers, such as type II collagen and Aggrecan, and reduce the expression of chondrocyte catabolism mediators and cell death markers. MT can also promote the expression of TGF-β1 in chondrocytes and induce the synthesis of ECM [51,52]. Daily oral administration or intraperitoneal injection of MT has been reported to improve bone deformity and osteolysis [53,54]. In in vivo experiments, MT showed a positive effect in alleviating the progression of OA and improving bone quality.

IHC, WB, and qRT-PCR analyses indicated that MT could reduce the expression of chondrocyte matrix degradation-related proteins MMP-3, MMP-13, and ADAMTS-4, and increase the expression levels of cartilage repair components type II collagen and Aggrecan. ADAMTS-4 and MMP-3 are ECM hydrolases that are important markers of joint degeneration [55]. Zhang Yijian found that MT promoted the expression levels of cartilage matrix synthesis markers SRY-box transcription factor 9 (SOX-9) and type II collagen, while reducing cartilage matrix degradation proteins [56]. We also found that the expression levels of iNOS and COX-2 in IL-1β-induced chondrocytes significantly decreased after MT treatment. Excessive iNOS catalyzes the production of a large amount of NO, thereby inhibiting the synthesis of proteoglycan and collagen, activating metalloproteinases, inducing apoptosis, and regulating inflammatory responses. Multiple experiments have shown that MT can inhibit the expression of ROS in cartilage, reduce the degradation of type II collagen and proteoglycans, restore cartilage matrix and chondrogenic gene expression and promote cell proliferation [27,57,58]. MT may inhibit the expression levels of chondrocyte matrix degradation-related proteins by inhibiting oxidase overexpression. However, the specific mechanism of action is still unclear and needs further exploration.

To explore the mechanism of MT in OA, we further pursued the matter by means of in vitro experiments. The results of CCK-8 showed that the co-culture of MT and IL-1β had no toxic effect on chondrocytes and could promote the proliferation of chondrocytes. IL-1β could induce inflammation by activating multiple pathways and could also promote the secretion of MMPs and pro-inflammatory mediators to accelerate cartilage destruction [59]. Therefore, in our in vitro experiments, we used IL-1β to mimic the OA environment. SIRT1 plays a role in the fight against OA by protecting chondrocytes through various pathways, such as reducing oxidative stress and the degree of inflammation [60,61]. SIRT1 is essential for maintaining cartilage integrity throughout life, and SIRT1 deficiency activates p53/p21-mediated senescence-associated secretory phenotypes, hypertrophy, and apoptosis and accelerates the onset of OA [62,63]. Conversely, MT can inhibit the production of MMP-3 and MMP-13 by inhibiting the expression and activity of SIRT1 in chondrocytes stimulated by IL-1β [20]. According to Ye dong’s findings, MT treatment inhibits SIRT1 exerting antitumor activity in human osteosarcoma cells, and the results suggested that MT is an inhibitor targeting SIRT1 signaling [64]. There is still controversy about the role of MT on SIRT1, and different experiments have shown different results. To validate the relationship between MT and SIRT1 in IL-1β-induced rat chondrocytes, we introduced EX527, a known inhibitor of SIRT1. Our study found that IL-1β showed the same trend as EX527, and both could reduce the expression of SIRT1. MT could restore the expression of SIRT1 reduced by IL-1β but could not restore the level of SIRT1 reduced by the inhibitor. The introduction of SIRT1 inhibitors further confirmed the activation of SIRT1 by MT in IL-1β-induced chondrocytes.

As one of the important factors of the inflammatory immune response, NF-κB plays an important role in the pathogenesis of OA. P I Sidiropoulos et al. have suggested that the activation of the NF-κB signaling pathway can initiate the release of inflammatory factors and MMPs, and accelerate the degree of OA [65]. Based on previous literature [19,66], the protective effect of MT on chondrocytes might be related to the NF-κB signaling pathway. WB results showed that the protein expression levels of p-p65, p-IκBα, and IκBα returned to normal levels after MT treatment. MT may protect chondrocytes by inhibiting the activation of the NF-κB signaling pathway to effectively inhibit the expression levels of chondrocyte matrix degradation-related proteins. Peiqiang Su et al. have suggested that MT may reverse IL-1β-induced impaired chondrogenic differentiation in BMSCs by attenuating activation of the NF-κB signaling pathway [67]. Several studies have found that targeting increased SIRT1 activity can inhibit NF-κB activation, reducing inflammation and oxidative stress in articular cartilage [35,68]. To examine the relationship between MT and the SIRT1/NF-κB pathway, we introduced the SIRT1 inhibitor EX527. We found that EX527 exhibited the same trend as IL-1β. The addition of EX527 reversed the inhibitory effect of MT on various indicators in the NF-κB pathway. Zhou Long found that MT promoted the expression of SIRT1 protein, and adding an MT receptor antagonist or SIRT1 antagonist reversed the anti-aging effect of MT [69]. Many results are consistent with our findings. MT inhibits the overexpression of the NF-κB signaling pathway by activating SIRT1 in IL-1β- treated chondrocytes, thereby inhibiting the expression of downstream indicators of the NF-κB pathway.

In the underlying study, MT treatment significantly promoted the expression of type II collagen and the transcription level of mRNA in IL-1β-induced chondrocytes. Type II collagen is synthesized by chondrocytes and secreted into the ECM to protect the chondrocytes. Therefore, the up-regulation of type II collagen expression may be related to MT’s ability to promote cartilage repair. In articular cartilage, high expression of TGF-β1 plays an important role in cartilage ECM synthesis, differentiation, adhesion, and migration [70,71,72]. Huang Chong et al. found that intra-articular injection of MT solution at a concentration of 20 mg/mL could inhibit the development of OA, which might be related to the up-regulation of the expression of cartilage growth factor TGF-β [21]. The up-regulation of collagen type II in chondrocytes might be related to the TGF-β pathway. Through WB, we found that MT could activate the TGF-β1/Smad2 pathway. However, the addition of EX527 could not cancel the effect of MT on these indicators, which indicated that the stimulatory effect of MT on the TGF-β1/Smad2 pathway was not mediated by SIRT1. Upregulation of SIRT1 inhibited TGF-β1/Smads signaling in hepatocytes and nephropathic epithelial cells [73,74]. However, in chondrocytes, there is no literature on the relationship between SIRT1 and TGF-β1/Smad2. In addition, the up-regulation of type II collagen expression by MT might be related to the activation of the TGF-β1 pathway. However, the addition of EX527 inhibited the expression of type II collagen in this study, which suggested that the repairing effect of activating the TGF-β1/Smad2 pathway was much less than the damaging effect of inhibiting SIRT1.

Through WB, qRT-PCR, and IF analyses, we found that EX527 up-regulated the expression levels of cell-matrix degradation-related proteins MMP3, MMP13, ADAMTS-4, iNOS, and COX-2 and decreased the expression of type II collagen. The addition of MT did not change this result, which suggested that inhibition of SIRT1 increased the risk of chondrocyte matrix degradation. The expression of SIRT1 in IL-1β-stimulated chondrocytes was reduced, and MT could reduce the expression of chondrocyte matrix degradation-related proteins by up-regulating the expression of SIRT1, thereby protecting chondrocytes.

MT protects chondrocytes by promoting the synthesis and secretion of type II collagen and reducing the degradation of ECM. This effect may be achieved by activating the TGF-β1/Smad2 pathway. On the other hand, MT can inhibit the expression of chondrocyte matrix degradation-related proteins by activating SIRT1 to inhibit the NF-κB pathway to reduce chondrocyte damage (Figure 9). MT has become well-known in recent years because of its many functions and has been used in various scientific research fields. It has also become a research hotspot in the direction of OA. The combined application of MT and material science also show a protective effect on chondrocytes. However, the protective mechanism of MT in OA remains to be further studied [49,75]. In this study, we discovered part of the mechanism by which MT protects chondrocytes and provided new theoretical support for the treatment of OA.

Although a lot of research has been done in this study, there are still many shortcomings. We found that inhibition of SIRT1 did not affect the activation of TGF-β1/Smad2 by MT, but the lack of a SIRT1 activator further verified this effect. In addition, the relationship between the NF-κB and TGF-β1/Smad2 pathways has not been thoroughly explored. In future studies, we should further explore the relationship between MT and the TGF-β signaling pathway and the protective effect of MT on OA chondrocytes. The introduction of SIRT1 activator resveratrol verified the positive effect of MT on the TGF-β pathway. It provides new theoretical support for its mechanism of delaying the progression of OA and reducing the inflammatory response.

## 5. Conclusions

MT can protect chondrocytes by activating the TGF-β1/Smad2 pathway and promoting the expression of type II collagen. It can also inhibit the expression of chondrocyte matrix degradation-related proteins by activating SIRT1 to inhibit the NF-κB pathway (Figure 9).

## Figures and Tables

**Figure 1 nutrients-14-03966-f001:**
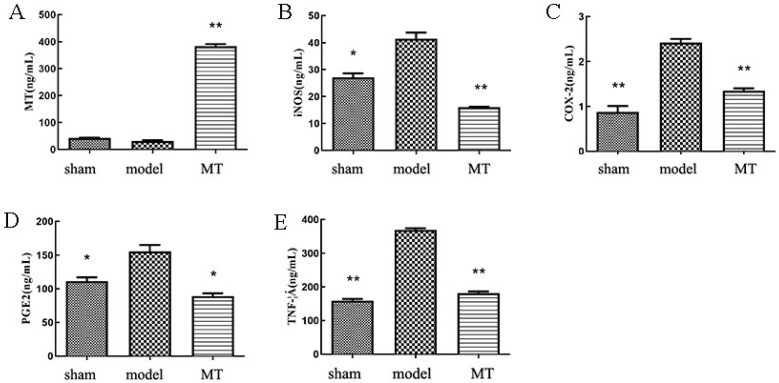
The circulating levels of melatonin (MT) and inflammatory factors in the serum of rats in each group. (**A**–**E**) Enzyme-linked immunosorbent assay (ELISA) kits were used to detect the circulating levels of MT, inducible nitric oxide synthase (iNOS), cyclooxygenase-2 (COX-2), prostaglandin E2 (PGE2), and tumor necrosis factor-alpha (TNF-α) in the serum of each group. All results represent mean ± standard deviation (SD) (*n* = 3), * *p* < 0.05, ** *p* < 0.01 (compared to the model group). MT group: 30 mg/kg/2d MT.

**Figure 2 nutrients-14-03966-f002:**
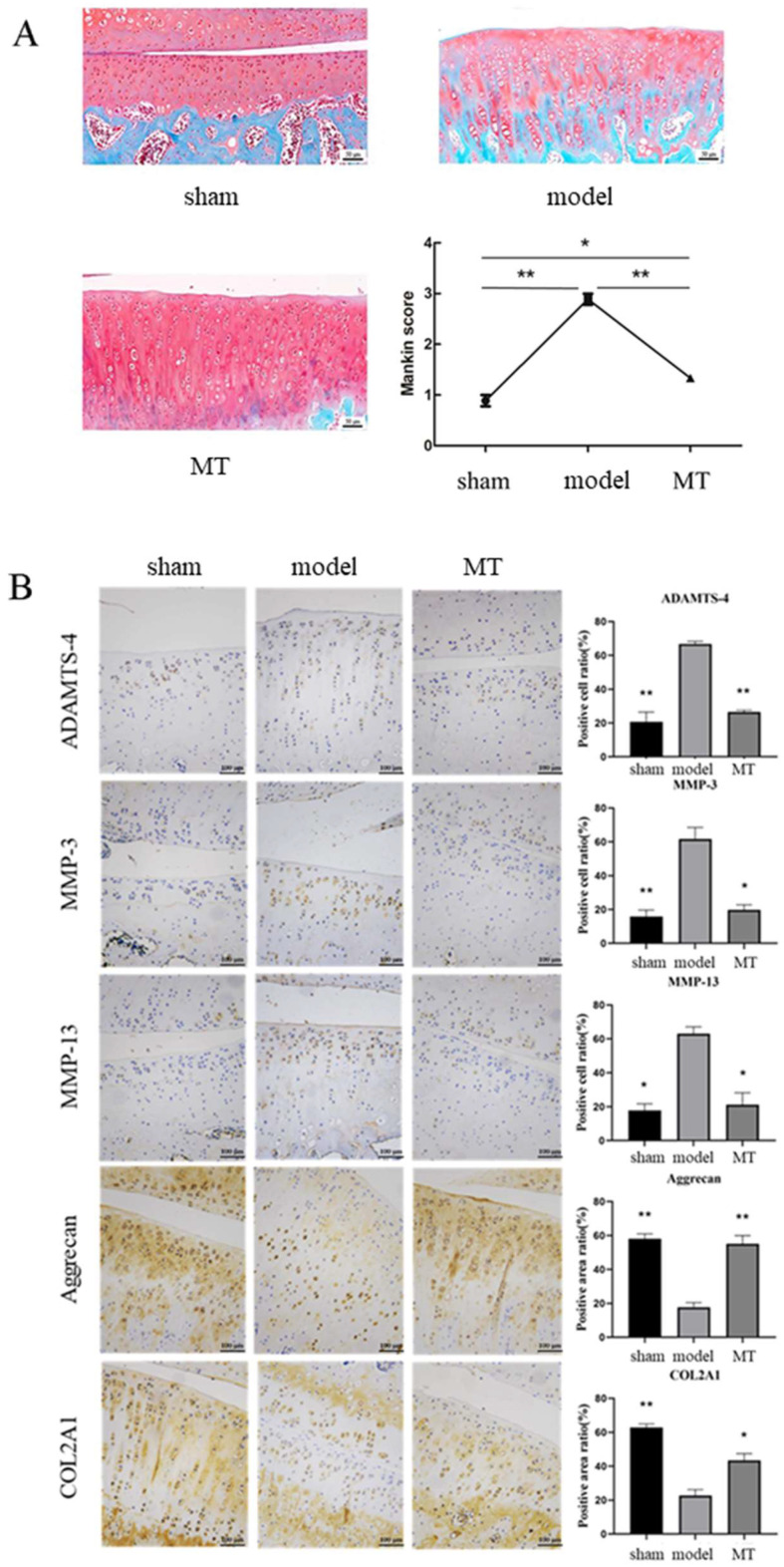
Melatonin (MT) inhibited chondrocyte matrix degradation in the rat anterior cruciate ligament transection (ACLT) model. (**A**) The surface of cartilage in each group was observed by Safranin O staining 12 weeks after the operation and scored by Mankin score. (**B**) Immunohistochemistry (IHC) staining and quantification of IHC staining for matrix metalloproteinase-3 (MMP-3), Aggrecan, matrix metalloproteinase-13 (MMP-13), ADAM metallopeptidase with thrombospondin type 1 motif 4 (ADAMTS-4) and collagen type II. * *p* < 0.05, ** *p* < 0.01 (compared to the model group).

**Figure 3 nutrients-14-03966-f003:**
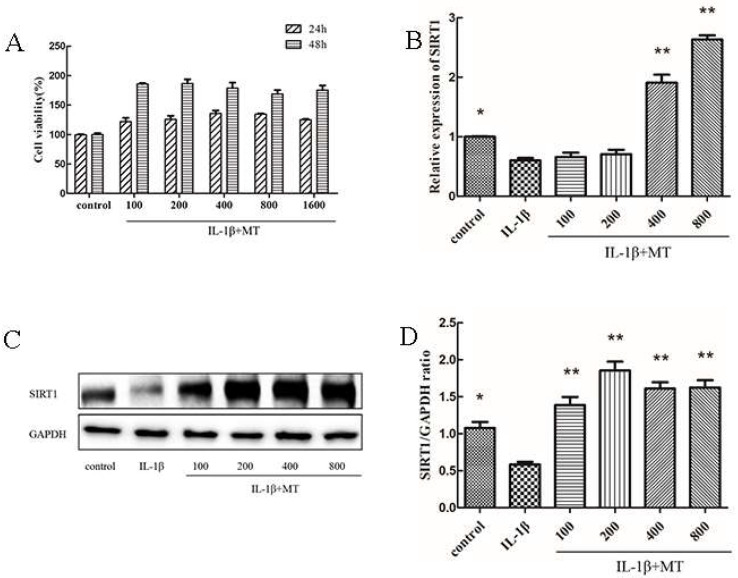
Melatonin (MT) increased chondrocyte viability and up-regulated sirtuin 2-related enzyme 1 (SIRT1) expression in interleukin 1-beta (IL-1β)-treated chondrocytes. (**A**) The effect of MT on the activity of rat chondrocytes for 24 and 48 h. (**B**) The transcription level of SIRT1 messenger RNA (mRNA) was detected by real-time PCR (qRT-PCR). (**C**,**D**) Western blot (WB) to detect the expression level of SIRT1 protein. Gray values were analyzed with ImageJ. All values represent mean ± standard deviation (SD) (*n* = 3), * *p* < 0.05, ** *p* < 0.01 (compared with the model group).

**Figure 4 nutrients-14-03966-f004:**
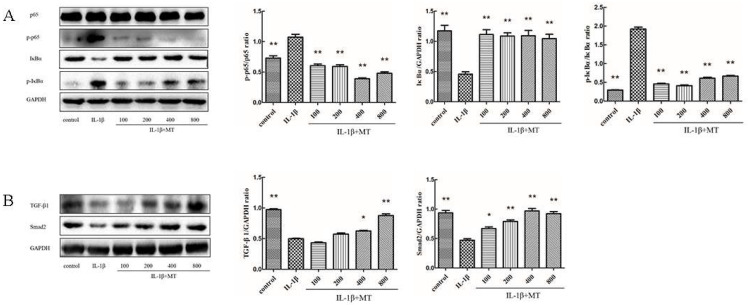
Melatonin (MT) inhibited the nuclear factor κB (NF-κB) signaling and activated the transforming growth factor-beta 1 (TGF-β1)/SMAD family member 2 (Smad2) pathways in interleukin 1-beta (IL-1β)-treated chondrocytes. (**A**,**B**) Western blot (WB) was used to detect protein expression levels of p65, nuclear factor of kappa light polypeptide gene enhancer in B-cells inhibitor (p-p65), alpha (IκBα), p-IκBα, TGFβ1, and Smad2. * *p* < 0.05, ** *p* < 0.01 (compared to the model group).

**Figure 5 nutrients-14-03966-f005:**
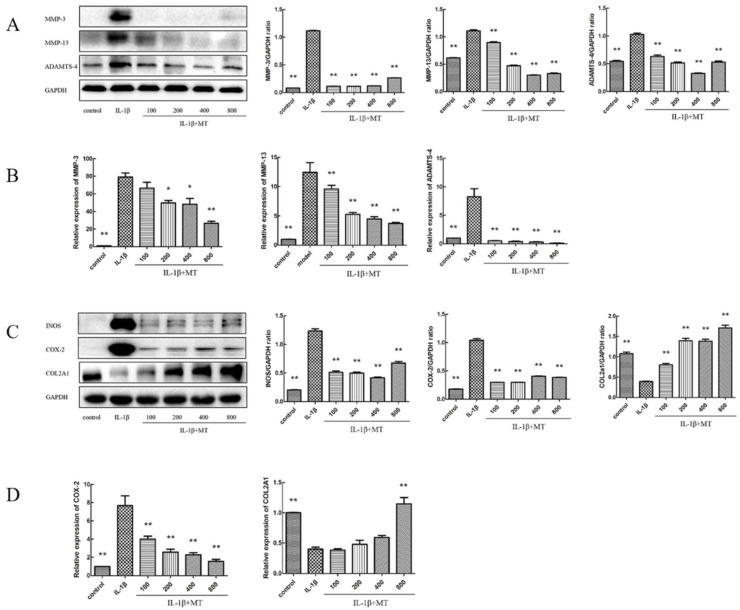
Melatonin (MT) inhibited the degradation of the interleukin 1-beta (IL-1β)-induced chondrocyte extracellular matrix. (**A**,**C**) Western blot (WB) detected the expression levels of matrix metalloproteinase-3 (MMP-3), matrix metalloproteinase-13 (MMP-13), ADAM metallopeptidase with thrombospondin type 1 motif 4 (ADAMTS-4), inducible nitric oxide synthase (iNOS), cyclooxygenase-2 (COX-2), and collagen II. (**B**,**D**) Real-time PCR (qRT-PCR) detected the transcript levels of MMP-3, MMP-13, ADAMTS-4, COX-2, and collagen II. * *p* < 0.05, ** *p* < 0.01 (compared to the model group).

**Figure 6 nutrients-14-03966-f006:**
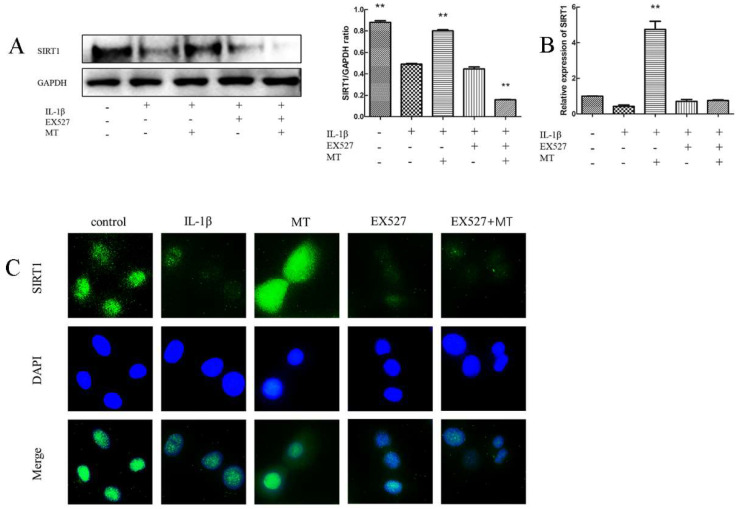
EX527 abrogated melatonin (MT)-induced sirtuin 2-related enzyme 1 (SIRT1) activation in interleukin 1-beta (IL-1β)-treat chondrocytes. (**A**) Western blot (WB) was used to detect the expression level of SIRT1 protein. (**B**) Transcription levels of SIRT1 messenger RNA (mRNA) were detected by real-time PCR (qRT-PCR). (**C**) Immunofluorescence (IF) staining was used to detect the number of SIRT1-positive chondrocytes in each group. ** *p* < 0.01 (compared to the model group). EX527: Classic SIRT1 inhibitor; DAPI: 4’,6-diamino-2-phenylindole.

**Figure 7 nutrients-14-03966-f007:**
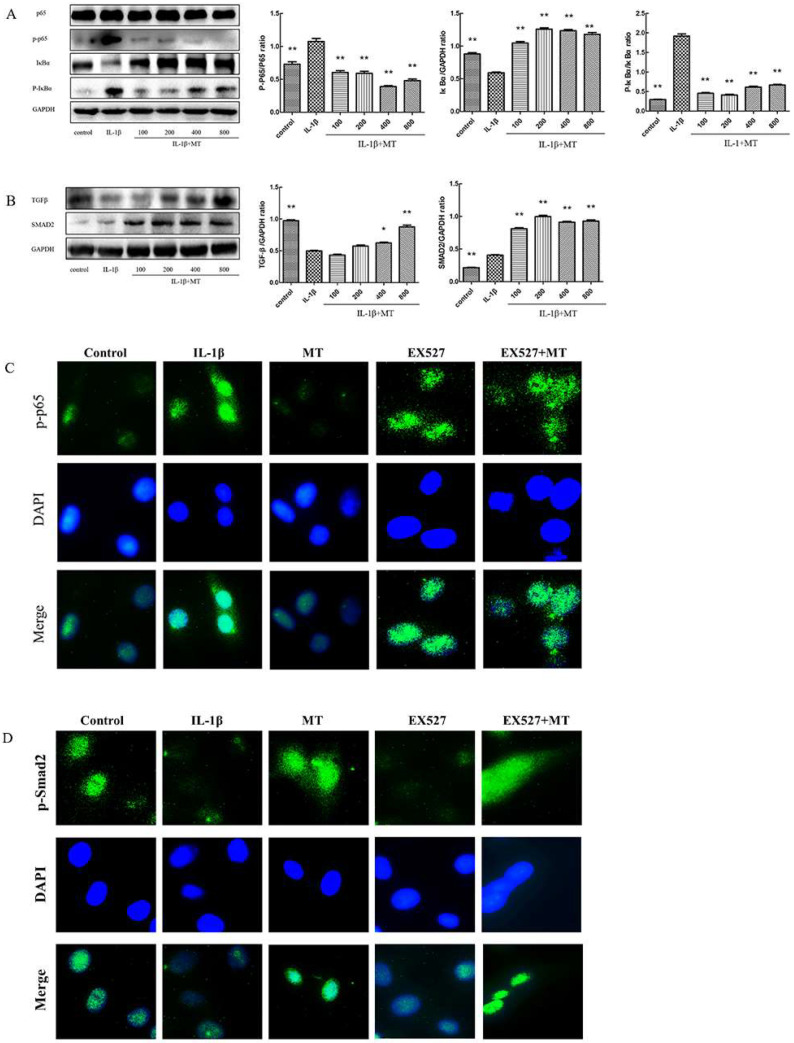
Effects of EX527 on the nuclear factor kappa B (NF-κB) and transforming growth factor beta (TGF-β)/SMAD family member 2 (Smad2) pathways in melatonin (MT)-treated interleukin 1-beta (IL-1β)-treated chondrocytes. (**A**,**B**) Western blot (WB) was used to detect the protein expression levels of p65, nuclear factor of kappa light polypeptide gene enhancer in B-cells inhibitor(p-p65), alpha (IκBα), p-IκBα, transforming growth factor beta (TGF-β1), and Smad family member 2 (Smad2). (**C**,**D**) Immunofluorescence (IF) staining was used to check the nuclear entry of p65 and Smad2 in each group. * *p* < 0.05, ** *p* < 0.01 (compared to the model group).

**Figure 8 nutrients-14-03966-f008:**
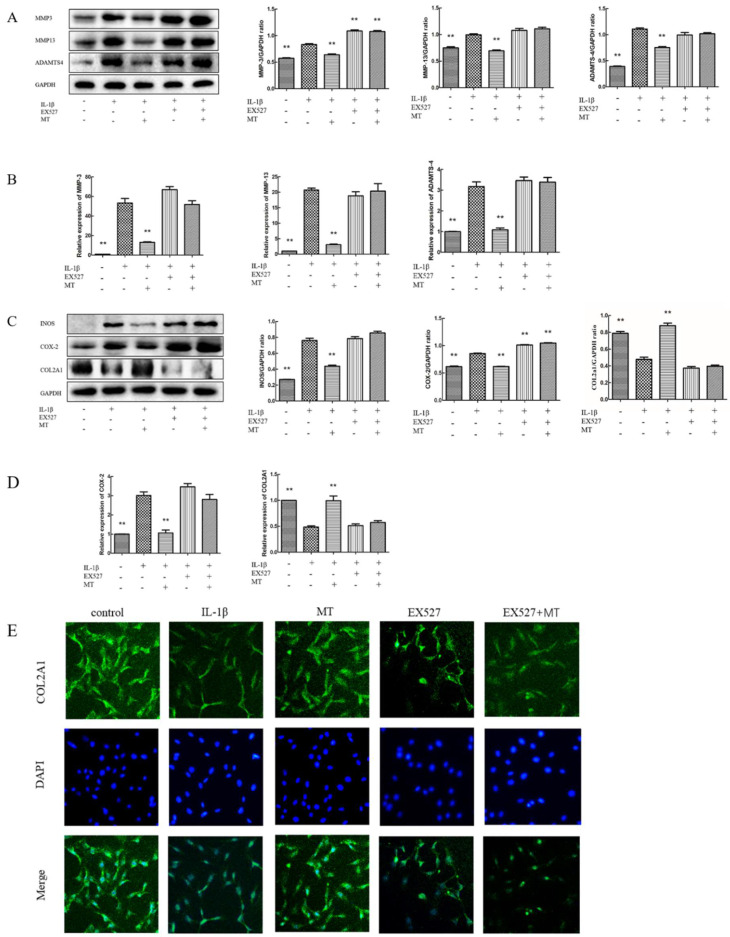
EX527 abrogated the inhibitory influence of MT on IL-1β-induced chondrocyte matrix degradation. (**A**,**C**) Western blot (WB) was used to detect the levels of matrix metalloproteinase-3 (MMP-3), matrix metalloproteinase-13 (MMP-13), ADAM metallopeptidase with thrombospondin type 1 motif 4 (ADAMTS-4), inducible nitric oxide synthase (iNOS), cyclooxygenase-2 (COX-2), and type II collagen. (**B**,**D**) Real-time PCR (qRT-PCR) was used to detect the levels of MMP-3, MMP-13, ADAMTS-4, COX-2, and type II collagen. (**E**) The intensity of collagen type II alpha 1 chain (COL2A1) in each group was detected by IF staining. ** *p* < 0.01 (compared to the model group).

**Figure 9 nutrients-14-03966-f009:**
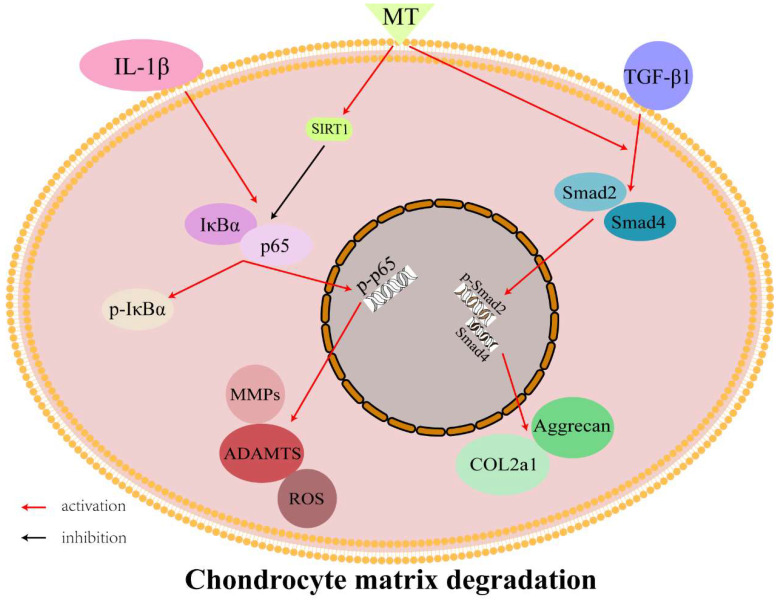
Interaction between melatonin (MT)/sirtuin 2-related enzyme 1 (SIRT1)/nuclear factor kappa B (NF-κB)/transforming growth factor-beta 1 (TGF-β1)/SMAD family member 2 (Smad2).

**Table 1 nutrients-14-03966-t001:** Primer sequence.

	Forward Primer	Reverse Primer
SIRT1 MMP-3	5′-TCAGTGTCATGGTTCCTTTGC-3′ 5′-TTTGGCCGTCTTCTCATCC-3′	5′-AATCTGCTCCTTTGCCACTCT-3′ 5′-GCATCGATCTTCTGGACGGT-3′
MMP-13	5′-TTCTGGTCTTCTGCCACACG-3	5′-TGGAGCTGCTTGTCCAGGT-3′
ADMTS-4	5′-AGGAGGCGCCCTTAACTCTG-3′	5′-CTACTCAGCGAAGCGAAGCG-3′
COX-2	5′-AGAAGCGAGGACCTGGGTTCAC-3′	5′-ACACCTCTCCACCGATGACCTG-3′
COL2A1	5-′ACGAAGCGGCTGGCAACCTCA-3′	5′-CCCTGTGAATGGGCGGAAAG-3′
GAPDH	5-’GATGCCCCCATGTTTGTGAT-3′	5-′GGCATGGACTGTGGTCATGAG-3′

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
