# Peer review of "Melatonin Prevents Chondrocyte Matrix Degradation in Rats with Experimentally Induced Osteoarthritis by Inhibiting Nuclear Factor-κB via SIRT1"

_nutrients, 2022, doi:10.3390/nu14193966_

Round 1
Reviewer 1 Report
Thank you for the opportunity to review this manuscript, which explored the in vitro and in vivo influence of MT on OA progression in rats.
Abstract:
- It would be helpful to provide the route of administration of MT in this section.
Introduction:
- It would be helpful to more consistently provide the following when citing previous studies that considered MT and OA: species utilized, in vitro versus in vivo work, and method of MT administration.
Materials and Methods:
- Upon what was the dose, frequency, and route of administration of the MT based? Pilot work by the authors and/or previous work? Also, in what was the MT dissolved (if it was)? If a vehicle was utilized to solubilize the MT, this should have been included as a control group. Finally, is it known whether IP administration of MT results in intra-articular presence?
- Reference and a brief description of the Mankin score for OA that was used needs to be provided. Also, what plane of section was evaluated?
- Description of the IHC processes do not permit a reader to recapitulate the experiments. Further, there is no mention of a negative control, which is needed to appropriately quantitate immunostaining and account for background staining. Indeed, it was a bit surprising the ADAMTS4 would be found in the ECM of cartilage?
- Lines 159-165: There is no mention in this section as to whether the influence of a vehicle control is needed for these cell culture experiments? The authors indicate that the control group DMEM-F12 with FBS; however, it is not clear that this is the condition used when treating cells with the IL-1B and/or MT? Further, what is the influence of MT, alone, on chondrocytes? In this same vein, what does the inhibitor EX527 do in isolation to chondrocytes? It is also not clear how cell viability and mitochondrial activity can be calculated by dehydrogenase activity?
- Line 189: This method should be provided as "qRT-PCR." Also, was a housekeeping gene utilized?
Results:
- Lines 220-222: Use of "secretion" here should be replaced by "increased circulating levels of..." For the corresponding Figure Legend, use of "contents" should also be replaced.
- Lines 229-238: It is kindly suggested that more appropriate histopathologic terminology be used in this section. Also, the photos provided in Figure 2A do not appear to be at the same location within the joints; this should be corrected. These images are also quite small and hard to evaluate.
- Figure 2B: As mentioned for Figure 2A, equivalent regions/areas need to be evaluated in order to compare immunostaining. Further, it is kindly suggested that images look as similarly oriented as possible with equivalent white balance.
- Figure 3A: The authors indicate that this data suggests that cell viability was not decreased by IL-1B; however, no mention seems to be made about an apparent increased response (presumably by dehydrogenase activity)?
Discussion:
- Further evaluation of this section requires addressing concerns raised above.
- One topic that would benefit this section would be to address the influence of MT at the dose utilized may have on the health and behavior of these rats (both of which can influence a multifaceted disease such as OA). In the absence of knowing what the MT did to these animals on a systemic level, a direct conclusion (versus an indirect influence) between MT and the in vivo results may be misleading.
Conclusion:
- Use of "repair" may not be the most appropriate choice here.
Author Response
I use red markers to indicate modifications to your proposal. The green part is the revision comments of another reviewer. The yellow part is the modification to reduce the repetition rate and language polish.
Abstract:
- It would be helpful to provide the route of administration of MT in this section.
I have added the route of administration to the abstract: section: In vivo experiment: intraperitoneal injection,In vitro experiment: co-culture of melatonin and culture medium
Introduction:
- It would be helpful to more consistently provide the following when citing previous studies that considered MT and OA: species utilized, in vitro versus in vivo work, and method of MT administration.
I have added in the introduction section the species used for melatonin, in vitro, in vivo and MT administration methods
Materials and Methods:
- Upon what was the dose, frequency, and route of administration of the MT based? Pilot work by the authors and/or previous work? Also, in what was the MT dissolved (if it was)? If a vehicle was utilized to solubilize the MT, this should have been included as a control group. Finally, is it known whether IP administration of MT results in intra-articular presence?
In vivo experiment: intraperitoneal injection, 30mg/Kg, once every two days; In vitro experiment: co-culture with melatonin, 10 ng/ml IL-1β and culture medium (DMEM-F12+10% FBS), 100, 200, 400, 800 ng/ml for 24 hours
According to the preliminary experiments, we found that melatonin has no toxic effect on chondrocytes, so we chose a relatively low concentration for administration.
Melatonin was dissolved in DMSO and then diluted with culture medium (DMEM-F12+10%FBS)。For almost all cells, DMSO concentrations of less than 0.1% are safe. In this experiment, the concentration of DMSO was much less than 0.1%, so it was not used as a control group.
According to previous literature, melatonin receptors have been found in cartilage. Production of melatonin was confirmed by mass spectrometric analysis of primary rat and chick chondrocytes. These data indicate that chondrocytes produce melatonin, which regulates cartilage growth and maturation via the MT1 and MT2 receptors. the melatonin receptors Mt1 and Mt2 in peripheral regions of cartilage and hypertrophic cartilage in the growth plate, which are in close proximity to wellvascularized tissues. Experiments show that exogenous and endogenous melatonin act synergistically in chondrocytes to regulate rhythmic expression of the clock on the central optic chiasm. Melatonin also regulates cartilage regeneration and degradation by directly/indirectly regulating the expression of major circadian clock genes Melatonin attenuates the expression of TNF-α and nuclear IL-1β in synovial fibroblasts and reduces cartilage degradation. In addition, melatonin also reduces cartilage degradation by attenuating the expression of TNF-α and IL-1β in synovial fibroblasts.
[1] Fu S, Kuwahara M, Uchida Y, Koudo S, Hayashi D, Shimomura Y, Takagaki A, Nishida T, Maruyama Y, Ikegame M, Hattori A, Kubota S, Hattori T. Circadian production of melatonin in cartilage modifies rhythmic gene expression. J Endocrinol. 2019 Mar 1:JOE-19-0022.R2. doi: 10.1530/JOE-19-0022. Epub ahead of print. PMID: 30889551.
[2] Lu KH, Lu PW, Lu EW, Tang CH, Su SC, Lin CW, Yang SF. The potential remedy of melatonin on osteoarthritis. J Pineal Res. 2021 Oct;71(3):e12762. doi: 10.1111/jpi.12762. Epub 2021 Sep 5. PMID: 34435392.
[3] Huang CC, Chiou CH, Liu SC, Hu SL, Su CM, Tsai CH, Tang CH. Melatonin attenuates TNF-α and IL-1β expression in synovial fibroblasts and diminishes cartilage degradation: Implications for the treatment of rheumatoid arthritis. J Pineal Res. 2019 Apr;66(3):e12560. doi: 10.1111/jpi.12560. Epub 2019 Feb 14. PMID: 30648758.
- Reference and a brief description of the Mankin score for OA that was used needs to be provided. Also, what plane of section was evaluated?
I've added a reference and a brief description of the Mankin score
This experiment assessed the coronal plane of the articular cartilage.
The Mankin score is based on four aspects: the integrity of cartilage structure, the distribution of chondrocytes, the staining of cartilage matrix and the integrity of the tide line. The higher the score, the more severe the cartilage degeneration.
- Description of the IHC processes do not permit a reader to recapitulate the experiments. Further, there is no mention of a negative control, which is needed to appropriately quantitate immunostaining and account for background staining. Indeed, it was a bit surprising the ADAMTS4 would be found in the ECM of cartilage?
Many thanks to the reviewer for pointing out the flaws, I have added the detailed experimental steps of immunohistochemistry in the original text.
Many thanks to the reviewers for the revisions, from which I learned the importance of negative controls. The purpose of the negative control is to exclude false positives and is used to monitor some variables in the experiment. In vivo experiments, we used a very small amount of DMSO to dissolve melatonin and then diluted it to the target concentration with normal saline. Since less than 0.1% DMSO has almost no toxic effects, we regrettably did not add a negative control group.
Aggrecan is an important component of the extracellular matrix and is the main component of the extracellular matrix of normal cartilage. ADAMTS-4 disrupts the balance of extracellular matrix proteoglycan metabolism. Numerous studies have shown that ADAMTS4 and ADAMTS5 play a major role in cartilage degradation during inflammatory joint diseases such as osteoarthritis (OA).
- Lines 159-165: There is no mention in this section as to whether the influence of a vehicle control is needed for these cell culture experiments? The authors indicate that the control group DMEM-F12 with FBS; however, it is not clear that this is the condition used when treating cells with the IL-1B and/or MT? Further, what is the influence of MT, alone, on chondrocytes? In this same vein, what does the inhibitor EX527 do in isolation to chondrocytes? It is also not clear how cell viability and mitochondrial activity can be calculated by dehydrogenase activity?
Melatonin was dissolved in DMSO and then diluted with culture medium (DMEM-F12+10% FBS). A concentration of 0.1% DMSO is safe for almost all cells. In this experiment, the concentration of DMSO was much less than 0.1%, so it was not used as a control group.
DMEM-F12 and FBS are the conditions used when treating cells with IL-1B and/or MT.
Through preliminary experiments, we found that when the concentration of melatonin reaches 10mg/ml, it still has no toxic effect on chondrocytes, and it can promote the proliferation of chondrocytes.
According to the previous literature, we used the concentration of EX527 for reference, so this experiment did not detect the effect of EX527 on chondrocytes.
[1] Kang X, Yang W, Wang R, Xie T, Li H, Feng D, Jin X, Sun H, Wu S. Sirtuin-1 (SIRT1) stimulates growth-plate chondrogenesis by attenuating the PERK-eIF-2α-CHOP pathway in the unfolded protein response. J Biol Chem. 2018 Jun 1;293(22):8614-8625. doi: 10.1074/jbc.M117.809822. Epub 2018 Apr 13. PMID: 29653943; PMCID: PMC5986218.;
[2] Li J, Zhang Z, Qiu J, Huang X. 8-Methoxypsoralen has Anti-inflammatory and Antioxidant Roles in Osteoarthritis Through SIRT1/NF-κB Pathway. Front Pharmacol. 2021 Sep 6;12:692424. doi: 10.3389/fphar.2021.692424. PMID: 34552480; PMCID: PMC8450503.
- Line 189: This method should be provided as "qRT-PCR." Also, was a housekeeping gene utilized?
I have changed in the article to qRT-PCR
housekeeping gene GAPDH:
F:GATGCCCCCATGTTTGTGAT
R:GGCATGGACTGTGGTCATGAG
Results:
- Lines 220-222: Use of "secretion" here should be replaced by "increased circulating levels of..." For the corresponding Figure Legend, use of "contents" should also be replaced.
I have changed in the article to “increased circulating levels of..”
- Lines 229-238: It is kindly suggested that more appropriate histopathologic terminology be used in this section. Also, the photos provided in Figure 2A do not appear to be at the same location within the joints; this should be corrected. These images are also quite small and hard to evaluate.
I have changed to more appropriate histopathology terminology.
In order to see the pathological changes of cartilage tissue more clearly, we enlarged the stained pictures and selected the most typical parts of the joints to show their changes.
As per your suggestion, I have enlarged the stained picture.
- Figure 2B: As mentioned for Figure 2A, equivalent regions/areas need to be evaluated in order to compare immunostaining. Further, it is kindly suggested that images look as similarly oriented as possible with equivalent white balance.
I have followed your suggestion to make the pictures look similar and they have equivalent white balance.
- Figure 3A: The authors indicate that this data suggests that cell viability was not decreased by IL-1B; however, no mention seems to be made about an apparent increased response (presumably by dehydrogenase activity)?
Based on the CCK-8 results, we found that co-culture of melatonin with IL-1β promoted chondrocyte proliferation.
Discussion:
- Further evaluation of this section requires addressing concerns raised above.
- One topic that would benefit this section would be to address the influence of MT at the dose utilized may have on the health and behavior of these rats (both of which can influence a multifaceted disease such as OA). In the absence of knowing what the MT did to these animals on a systemic level, a direct conclusion (versus an indirect influence) between MT and the in vivo results may be misleading.
According to our monitoring of rat body weight, melatonin did not affect body weight gain. It can be seen that the dose of melatonin we used did not affect the feeding situation of the rats. Since this experiment did not design a behavioral experiment, we could not observe the changes in the behavior of rats, which was a loophole in our experiment, and we realized its importance to this experiment.
Melatonin is an important hormone that regulates circadian rhythms, and multiple studies have shown that a reduction in melatonin at systemic levels can lead to circadian rhythm disturbances. Disturbance of circadian rhythm is also one of the reasons for the induction of arthritis.
[1] Sumsuzzman DM, Choi J, Khan ZA, Kamenos G, Hong Y. Melatonin Maintains Anabolic-Catabolic Equilibrium and Regulates Circadian Rhythm During Osteoarthritis Development in Animal Models: A Systematic Review and Meta-analysis. Front Pharmacol. 2021 Sep 17;12:714974. doi: 10.3389/fphar.2021.714974. PMID: 34603028; PMCID: PMC8484877.
[2] Kouri VP, Olkkonen J, Kaivosoja E, Ainola M, Juhila J, Hovatta I, Konttinen YT, Mandelin J. Circadian timekeeping is disturbed in rheumatoid arthritis at molecular level. PLoS One. 2013;8(1):e54049. doi: 10.1371/journal.pone.0054049. Epub 2013 Jan 15. PMID: 23335987; PMCID: PMC3546002.
[3] Olkkonen J, Kouri VP, Hynninen J, Konttinen YT, Mandelin J. Differentially Expressed in Chondrocytes 2 (DEC2) Increases the Expression of IL-1β and Is Abundantly Present in Synovial Membrane in Rheumatoid Arthritis. PLoS One. 2015 Dec 28;10(12):e0145279. doi: 10.1371/journal.pone.0145279. PMID: 26710124; PMCID: PMC4692547.
Conclusion:
- Use of "repair" may not be the most appropriate choice here.
I have changed repair to protect
Reviewer 2 Report
The authors showed in rat chondrocyte monolayers that melatonin can protect against IL-1β-induced articular cartilage degradation by suppression of nuclear factor-κB via SIRT1. Besides this the TGF-β1/smad2 pathway was rescued, independent of SIRT1, but the protective effects were all regulated via SIRT1. They also showed that melatonin treatment protected articular cartilage in the ACLT model in rats. But the authors should not conclude from this that the described intracellular effects of melatonin occur in their osteoarthritis model, nor in human osteoarthritis, because there is no proof that IL-1 drives osteoarthritis.
For this reason, I propose another title “Melatonin protects against IL-1β-induced articular cartilage degradation by suppression of nuclear factor-κB via SIRT1.” This is the central finding and the reduction of degradative factors and upregulation of extracellular matrix molecules appeared to be dependent on SIRT1, and not on TGF-β1/smad2 signaling.
Further Comments:
Line 30: The authors mention activation of the TGF-β1/smad2 pathway first although its protective effects appeared to be subordinate to SIRT1, as mentioned above.
Line 32: The authors did not show SIRT1 activation, but SIRT1 levels.
Line 32: “SIRT-1”should be “SIRT1”
Line 33: “chondrocyte repair” should be “cartilage repair”. This same problem is found also in lines 64, 491,527, 541and 560.
Line 39-41: Very long sentence. I propose “OA is not easy to be diagnosed in the early stage. The main clinical symptoms are difficulty in movement caused by pain, and cartilage degeneration.”
Line 44: “problems such as age and cost have always existed” please explain better.
Line 45: do you mean “osteoarthritis”?
Lines 61-63: Here the authors mention an earlier study of Guo et al. (2017) showing in very comparable experiments that IL-1 increased SIRP1 levels in (human) chondrocytes and that melatonin suppressed this. Moreover, using the same in ACLT model they found that melatonin decreased SIRT1 protein expression. This is all opposite to the findings of the underlying study and this contradiction should get attention and be addressed in the Discussion section.
Line 66: I propose: “However, the role of SIRT1 still needs further experimental exploration.”
Line 78: The authors did not study the effect of melatonin on the NF-κB signaling pathway in the rat osteoarthritis model. They studied this in IL-1β- stimulated rat chondrocytes in monolayer, which is not osteoarthritis. Just remove this sentence.
Line 86: “Experiments found” should be “Earlier studies showed”
Line 90: “The purpose of this experiment” should be “The purpose of the underlying study”
Line 96: First write the section “Materials and Methods”
Line 97: Was this rat interleukin-1β? Or human or mouse?
Line 109: I propose “self-bred). Adult males (n=30, 300 ± 20g) were used….”
Line 120: amputation is not the same as transection.
Line 132: I propose: “…cartilage and bone. This was scored using an adapted version of the Mankin score (reference)
Lines 130- 131: How many sections were scored per animal and at which specific sites?
Line 144: this sentence is from the instruction manual.
Line 156: You cannot avoid this, but you can minimize it.
Lines 162 and 164: In the figure there is both 24h and 48h
Line 172: please say that EX527 is a SIRT1 inhibitor
Line 210: At least for the in vivo experiment this is not true. It was done once-only.
Line 216: should be “Results”
Line 247: Safranin O
Line 249: scored using the Mankin score
Line 254: in Figure #A the effects of melatonin and IL-1β alone are not shown, so you cannot say this
Line 256 : I would not call 10/ml mild.
Lines 254-257: You cannot say that incubation with IL-1β simulates osteoarthritis. The is no proof for an important role of IL-1 in osteoarthritis.
Line 288 : I propose: “We investigated whether melatonin could protect against extracellular matrix degradation in IL-1β-treated chondrocytes”
Line 293: should be “extracellular matrix”
Line 295: A,C) Western blot and B,D) qPCR detection of…
Lines 322 -323: another explanation may be that there is more IκBα, just because less of it was phosphorylated?
Line 326 “worked”?
Line 333: “the role of Smad2 into the nucleus” should be “the translocation of Smad2 into the nucleus”
Line 333: I would add to “does not have this effect”, indicating that this was independent of SIRT1
Line 346: EX527 should be IL-1β+EX527
Lines 359-360: Western blot and qPCR were used to detect the levels of A,B) MMP3, MMP-13 and ADAMNTS-4 and C,D) iNOS, COX-2 and type II collagen
Figure 8C: Col2A protein expression is missing
Line 363: experimental osteoarthritis
Line 428: “…but the specific matrix is still unclear and…” I do not follow
Line 435: Again: It has never been proven that IL-1β is an important factor in inducing arthritis!
Line 463: “Based on previous literature” please provide references.
Line 472: 8-MOP this molecule is not introduced, what is it?
Line 486: In the underlying study
Line 506: “various indicators” should be “these indicators”
Line 508: ferndiene is not introduced, what is it
Line 524: “osteoarthritis chondrocytes “should be “IL-1β-stimulated chondrocytes”
Line 541: “and relieve pain” The authors did not study pain.
Lines 547-548 “between the two” which two?, please make clear.”
Line 549: “But the lack of SIRT1 activator further verified this” Did the authors study this?
Line 554 “SIRT1 activator resveratrol”this is a strange sentence, because this was not used in this study
Line 557: pain was not studied here
Conclusion: In rat chondrocyte monolayers melatonin can protect against IL-1β-induced articular cartilage degradation by suppression of nuclear factor-κB via SIRT1. Besides this the TGF-β1/smad2 pathway was rescued, independent of SIRT1, but the protective effects were all regulated via SIRT1. Melatonin treatment protected articular cartilage in the ACLT model in rats, possibly via the same mechanisms.
To my opinion, the Discussion section should be shortened.
Author Response
I marked your revision in green. The red part is the revision opinion of another reviewer. The yellow part is the modification to reduce the repetition rate and language polish.
The authors showed in rat chondrocyte monolayers that melatonin can protect against IL-1β-induced articular cartilage degradation by suppression of nuclear factor-κB via SIRT1. Besides this the TGF-β1/smad2 pathway was rescued, independent of SIRT1, but the protective effects were all regulated via SIRT1. They also showed that melatonin treatment protected articular cartilage in the ACLT model in rats. But the authors should not conclude from this that the described intracellular effects of melatonin occur in their osteoarthritis model, nor in human osteoarthritis, because there is no proof that IL-1 drives osteoarthritis.
Many thanks to the reviewers for their clear logic in pointing out my inappropriate conclusions. I have modified this part in the article.
In addition, I will explain the role of IL-1β in osteoarthritis here. IL-1 plays an important role in the pathogenesis of osteoarthritis, and IL-1β plays a central role in the whole inflammatory process. After the combination of IL-1 and IL-1RII, it affects gene transcription and post-transcriptional modification in the nucleus mainly through the MAPK pathway and NFκB pathway, so that chondrocytes synthesize PGE2, cytokines, MO, MMP, proteoglycanase , causing joint inflammation and degradation of cartilage mechanisms. It has also been clearly demonstrated in many previous literatures that IL-1β can be used to mimic the osteoarthritis environment in vitro.
[1] Shao Junjie, Zhang Xianlong, Jiang Yao. The role of IL-1 in the pathogenesis of osteoarthritis and related gene therapy [J]. International Journal of Orthopedics,2007(05):309-311+322.
[2] Xu J, Ma X. Hsa_circ_0032131 knockdown inhibits osteoarthritis progression via the miR-502-5p/PRDX3 axis. Aging (Albany NY). 2021 May 25;13(11):15100-15113. doi: 10.18632/aging.203073. Epub 2021 May 25. PMID: 34032607; PMCID: PMC8221332.ï¼›
[3] Tan C, Zhang J, Chen W, Feng F, Yu C, Lu X, Lin R, Li Z, Huang Y, Zheng L, Huang M, Wu G. Inflammatory cytokines via up-regulation of aquaporins deteriorated the pathogenesis of early osteoarthritis. PLoS One. 2019 Aug 12;14(8):e0220846. doi: 10.1371/journal.pone.0220846. PMID: 31404098; PMCID: PMC6690536.ï¼›
[4] Zhu S, Deng Y, Gao H, Huang K, Nie Z. miR-877-5p alleviates chondrocyte dysfunction in osteoarthritis models via repressing FOXM1. J Gene Med. 2020 Nov;22(11):e3246. doi: 10.1002/jgm.3246. Epub 2020 Jul 14. PMID: 32584470.)
[5] Liao CR, Wang SN, Zhu SY, Wang YQ, Li ZZ, Liu ZY, Jiang WS, Chen JT, Wu Q. Advanced oxidation protein products increase TNF-α and IL-1β expression in chondrocytes via NADPH oxidase 4 and accelerate cartilage degeneration in osteoarthritis progression. Redox Biol. 2020 Jan;28:101306. doi: 10.1016/j.redox.2019.101306. Epub 2019 Aug 22. PMID: 31539804; PMCID: PMC6812020.
For this reason, I propose another title “Melatonin protects against IL-1β-induced articular cartilage degradation by suppression of nuclear factor-κB via SIRT1.” This is the central finding and the reduction of degradative factors and upregulation of extracellular matrix molecules appeared to be dependent on SIRT1, and not on TGF-β1/smad2 signaling.
Dear reviewer, thank you very much for your objection to the title, which I am not very satisfied with. The topic and reason you raised made me understand the mechanism of melatonin from another aspect.
Further Comments:
Line 30: The authors mention activation of the TGF-β1/smad2 pathway first although its protective effects appeared to be subordinate to SIRT1, as mentioned above.
Many thanks to the reviewers for pointing out ambiguities in this article. Due to the unclear language in the early stage, this part is ambiguous and misunderstood by readers. Activation of the TGF-β1/smad2 pathway is not subordinate to SIRT1, which I have now modified.
Line 32: The authors did not show SIRT1 activation, but SIRT1 levels.
Based on the reviewer's comments, I have changed to the more accurate expression "promoted expression levels of SIRT1"
Line 32: “SIRT-1”should be “SIRT1”
Changed SIRT-1 to SIRT1
Line 33: “chondrocyte repair” should be “cartilage repair”. This same problem is found also in lines 64, 491,527, 541and 560.
Changed "chondrocyte repair" to "cartilage repair"
Line 39-41: Very long sentence. I propose “OA is not easy to be diagnosed in the early stage. The main clinical symptoms are difficulty in movement caused by pain, and cartilage degeneration.”
I have accepted your change proposal and changed it in the article
Line 44: “problems such as age and cost have always existed” please explain better.
In joint replacement surgery, there is a problem with the service life of the artificial joint. As time goes by, the artificial joint may have problems such as wear and aging and cannot be used. It needs to be replaced with a new artificial joint. In addition, the current cost of joint replacement and artificial joint itself is relatively high.
Line 45: do you mean “osteoarthritis”?
Yes, it should be "Osteoarthritis"
Lines 61-63: Here the authors mention an earlier study of Guo et al. (2017) showing in very comparable experiments that IL-1 increased SIRP1 levels in (human) chondrocytes and that melatonin suppressed this. Moreover, using the same in ACLT model they found that melatonin decreased SIRT1 protein expression. This is all opposite to the findings of the underlying study and this contradiction should get attention and be addressed in the Discussion section.
Many thanks to the reviewers for their suggestions on this part. I've added an explanation of this part to the Discussion. The effect of melatonin on SIRT1 has been controversial, and different experiments have shown different results. This is also the fundamental reason why this experiment wants to explore the problem. We looked forward to exploring the true effect of melatonin on SIRT1 in IL-1β-induced rat chondrocytes. This study is an exploratory experiment, which clearly found that the expression of SIRT1 in IL-1β-induced chondrocytes was decreased, while melatonin treatment restored the expression of SIRT1 in IL-1β-induced chondrocytes. Different results may be caused by differences in the source and health of the cells, the concentration of melatonin and the treatment time.
Line 66: I propose: “However, the role of SIRT1 still needs further experimental exploration.”
I have accepted your suggestion and made changes in the text.
Line 78: The authors did not study the effect of melatonin on the NF-κB signaling pathway in the rat osteoarthritis model.
They studied this in IL-1β- stimulated rat chondrocytes in monolayer, which is not osteoarthritis. Just remove this sentence.
Line 86: “Experiments found” should be “Earlier studies showed”
I have accepted your suggestion and made changes in the text.
Line 90: “The purpose of this experiment” should be “The purpose of the underlying study”
I have accepted your suggestion and made changes in the text.
Line 96: First write the section “Materials and Methods”
I have accepted your suggestion and made changes in the text.
Line 97: Was this rat interleukin-1β? Or human or mouse?
Rat IL-1β
Line 109: I propose “self-bred). Adult males (n=30, 300 ± 20g) were used….”
I have accepted your suggestion and made changes in the text.
Line 120: amputation is not the same as transection.
I have accepted your suggestion and made changes in the text.
Line 132: I propose: “…cartilage and bone. This was scored using an adapted version of the Mankin score (reference)
I have accepted your suggestion and made changes in the text.
Lines 130- 131: How many sections were scored per animal and at which specific sites?
From each animal we obtained only the right hind limb knee joint and serum. Regarding the knee joint, we performed Safranin O-fast green staining and Immunohistochemistry (IHC) staining to observe the pathological changes of articular cartilage and the expression of matrix-related proteins. And it was evaluated by the Mankin score. For serum, we measured the levels of melatonin, COX-2, iNOS, TNF-α, and PGE2. in serum by ELISA. In addition, we also recorded the body weight data of the rats, but they were not presented in the article.
Line 144: this sentence is from the instruction manual.。
I have accepted your suggestion and made changes in the text.。
Line 156: You cannot avoid this, but you can minimize it.
I have accepted your suggestion and made changes in the text.
Lines 162 and 164: In the figure there is both 24h and 48h。
I have accepted your suggestion and made changes in the text.
Line 172: please say that EX527 is a SIRT1 inhibitor
I have accepted your suggestion and made changes in the text.
Line 210: At least for the in vivo experiment this is not true. It was done once-only.
I have accepted your suggestion and made changes in the text.
Line 216: should be “Results”
I have accepted your suggestion and made changes in the text.
Line 247: Safranin O
I have accepted your suggestion and made changes in the text.
Line 249: scored using the Mankin score
I have accepted your suggestion and made changes in the text.
Line 254: in Figure #A the effects of melatonin and IL-1β alone are not shown, so you cannot say this
I have accepted your suggestion and made changes in the text.
Line 256 : I would not call 10/ml mild.
I have accepted your suggestion and made changes in the text.
Lines 254-257: You cannot say that incubation with IL-1β simulates osteoarthritis. The is no proof for an important role of IL-1 in osteoarthritis.
Line 288 : I propose: “We investigated whether melatonin could protect against extracellular matrix degradation in IL-1β-treated chondrocytes”
I have accepted your suggestion and made changes in the text.
Line 293: should be “extracellular matrix”
I have accepted your suggestion and made changes in the text.
Line 295: A,C) Western blot and B,D) qPCR detection of…
I have accepted your suggestion and made changes in the text.
Lines 322 -323: another explanation may be that there is more IκBα, just because less of it was phosphorylated?
Normally, IκBα and p65 are bound together, and when the two dissociate, p-p65 and p-IκBα are generated. Therefore, we speculate that melatonin may reduce p-p65 and p-IκBα and increase the content of IκBα by preventing the dissociation process.
Line 326 “worked”?
I have accepted your suggestion and made changes in the text.
Line 333: “the role of Smad2 into the nucleus” should be “the translocation of Smad2 into the nucleus”
I have accepted your suggestion and made changes in the text.
Line 333: I would add to “does not have this effect”, indicating that this was independent of SIRT1.
I have accepted your suggestion and made changes in the text.
Line 346: EX527 should be IL-1β+EX527
I have accepted your suggestion and made changes in the text.
Lines 359-360: Western blot and qPCR were used to detect the levels of A,B) MMP3, MMP-13 and ADAMNTS-4 and C,D) iNOS, COX-2 and type II collagen
I have accepted your suggestion and made changes in the text.
Figure 8C: Col2A protein expression is missing
Thank you very much for pointing out the mistakes, I have added pictures to the text
Line 363: experimental osteoarthritis
I have accepted your suggestion and made changes in the text.
Line 428: “…but the specific matrix is still unclear and…” I do not follow
I have accepted your suggestion and made changes in the text.
Line 435: Again: It has never been proven that IL-1β is an important factor in inducing arthritis!
Line 463: “Based on previous literature” please provide references.
Thank you very much for pointing out the mistakes, I have added references to the text
Line 472: 8-MOP this molecule is not introduced, what is it
To shorten the discussion, I have consolidated several references from this section, and have now removed specific names. However, information and references on 8-MOP have been introduced below.
8-Methoxypsoralen (8-MOP) is a natural furanocoumarin with various biological activities 8-MOP has been found with multiple biological function, such as regulation of apoptosis and proliferation in tumors.
[1] Liu, Y., Zhang, G., Zeng, N., and Hu, S. (2017). Interaction between 8methoxypsoralen and Trypsin: Monitoring by Spectroscopic, Chemometrics and Molecular Docking Approaches. Spectrochim Acta A. Mol. Biomol.Spectrosc. 173, 188–195. doi:10.1016/j.saa.2016.09.015.
[2] 8-MOP is a photosensitizer, and it has been applied to immunotherapy (Hähnel, V., Weber, I., Tuemmler, S., Graf, B., Gruber, M., Burkhardt, R., et al.(2020). Matrix-dependent Absorption of 8-methoxypsoralen in Extracorporeal Photopheresis. Photochem. Photobiol. Sci. 19 (8), 1099–1103. doi:10.1039/ c9pp00378a.
Line 486: In the underlying study
I have accepted your suggestion and made changes in the text.
Line 506: “various indicators” should be I have accepted your suggestion and made changes in the text.“these indicators” ]
I have accepted your suggestion and made changes in the text.
Line 508: ferndiene is not introduced, what is it ferndiene.
Pterostilbene was written as ferndiene due to a misrepresentation. In order to shorten the discussion, I have consolidated several references from this section, and have removed specific names for now.
Line 524: “osteoarthritis chondrocytes “should be “IL-1β-stimulated chondrocytes”
I have accepted your suggestion and made changes in the text.
Line 541: “and relieve pain” The authors did not study pain.
I took your suggestion and removed this expression.
Lines 547-548 “between the two” which two?, please make clear.”
The controversial sentence has been removed.
Line 549: “But the lack of SIRT1 activator further verified this” Did the authors study this?
Line 554 “SIRT1 activator resveratrol”this is a strange sentence, because this was not used in this study SIRT1 activator resveratrol”
Line 557: pain was not studied here
The above three questions are jointly explained here. This part is to envisage the follow-up exploration of this experiment, and we hope that the follow-up experiments can provide more sufficient and powerful evidence for this experiment.
Conclusion: In rat chondrocyte monolayers melatonin can protect against IL-1β-induced articular cartilage degradation by suppression of nuclear factor-κB via SIRT1. Besides this the TGF-β1/smad2 pathway was rescued, independent of SIRT1, but the protective effects were all regulated via SIRT1. Melatonin treatment protected articular cartilage in the ACLT model in rats, possibly via the same mechanisms.
Change the conclusion to: In rat chondrocyte monolayers melatonin can protect against IL-1β-induced articular cartilage degradation by suppression of nuclear factor-κB via SIRT1. Besides this the TGF-β1/smad2 pathway was rescued, independent of SIRT1. Melatonin treatment protected articular cartilage in the ACLT model in rats, possibly via the same mechanisms
To my opinion, the Discussion section should be shortened.
I have accepted your suggestion to shorten the discussion section.
Round 2
Reviewer 1 Report
This reviewer appreciates the authors' efforts to address previously provided comments. Some of these are resolved; however, some also remain unresolved.
Title: The current title attempts to conjoin the in vitro and in vivo work in a way that is perhaps not quite accurate.
Introduction:
- It is suggested that references be provided for the first paragraph of this section, which focuses on OA.
- Lines 49-50: This sentence would benefit from having a reference.
- Lines 52-53: The phrase "... and protects cartilage in various ways, such as intra-articular and intraperitoneal injections" is perhaps not quite correct and does not flow well. Perhaps a better option is making this a separate sentence: "Efficacy has been demonstrated following both intra-articular and intra-peritoneal injections" with associated references.
- Lines 55-67: This section would benefit from being more clear as to what is not known about MT-related SIRT1 regulation. As it stands, the authors provide a lot of information without leading the reader through how their particular project attempts to clarify this knowledge gap.
- Lines 68-73: References should be provided for this information. Again, for this section, it is not quite clear what the authors' current work does to add to this knowledge base.
Materials and Methods:
- Respectfully, the authors response to the initial comment related to justification of the dose, frequency, and route of administration of MT in either the responses or the updated manuscript is not adequately provided. For example, if one of the references provided in the response provides justification, this is not clear nor is it added to the manuscript.
- The IHC methods provided read as a protocol and are not appropriate language for a manuscript. Further, while the authors acknowledged the importance of negative controls, they did not provide the requested information related to use of such, which is critical for evaluating the results in this section.
- Lines 182-184: While it is appreciated that the concentration of DMSO utilized for both the in vivo and in vitro work may be considered "safe" or "have no cytotoxic effects," this does not negate its importance as being more thoroughly considered as a negative control. It is also not clear if the model group received any IP injections?
Results:
- It is appreciated that improved images are provided for Figure 2. However, the language utilized to describe the findings for the model group do not match the Saf-O/Fast Green images. In addition, the areas evaluated in the 3 groups are at different regions within the joint. Measure bars should also be provided on the photomicrographs.
- It still remains a concern that the absence of specific control groups limits the interpretation of the in vitro findings. At a minimum, this would mean that the language surrounding interpretations should be less definitive and this limitation should be provided in the Discussion.
Author Response
Many thanks to the reviewers for their suggestions. Your suggestions fill in many of the holes I left in the writing process. Your guidance has made my article more complete and rigorous. For your suggestions, I have responded one by one and modified them using revision mode in the article.
Title: The current title attempts to conjoin the in vitro and in vivo work in a way that is perhaps not quite accurate.
I changed the title to“Melatonin prevents chondrocyte matrix degradation in experimentally induced osteoarthritis rats by inhibiting nuclear factor-κB via SIRT1”
Introduction:
- It is suggested that references be provided for the first paragraph of this section, which focuses on OA.
I have added references to OA
- Lines 49-50: This sentence would benefit from having a reference.
I have added relevant references
- Lines 52-53: The phrase "... and protects cartilage in various ways, such as intra-articular and intraperitoneal injections" is perhaps not quite correct and does not flow well. Perhaps a better option is making this a separate sentence: "Efficacy has been demonstrated following both intra-articular and intra-peritoneal injections" with associated references.
I have changed this sentence to“Animal models such as rats and pigs and in vitro experiments have confirmed that MT plays an important role in various ways such as intra-articular injection and intraperitoneal injection. In these models, melatonin can maintain the activity of chondrocytes, pro-mote the synthesis of articular chondrocyte matrix, and protect articular cartilage and chondrocytes.”References 13-16
- Lines 55-67: This section would benefit from being more clear as to what is not known about MT-related SIRT1 regulation. As it stands, the authors provide a lot of information without leading the reader through how their particular project attempts to clarify this knowledge gap.
I added a sentence to illustrate the purpose of this experiment. We sought to explore the relationship between MT and SIRT1 in IL-1β-induced chondrocytes:It can be seen that previous studies have found that there is an interaction relationship between MT and SIRT1, but there are opposite experimental results. Therefore, this experiment is expected to explore the mechanism of MT and SIRT1 in IL-1β-induced chondrocytes.
- Lines 68-73: References should be provided for this information. Again, for this section, it is not quite clear what the authors' current work does to add to this knowledge base.
I have added references. And added a sentence to indicate the purpose of this experiment: Our study aimed to explore the interaction between the MT/SIRT1/NF-κB signaling pathway in IL-1β-induced chondrocytes.
Materials and Methods:
- Respectfully, the authors response to the initial comment related to justification of the dose, frequency, and route of administration of MT in either the responses or the updated manuscript is not adequately provided. For example, if one of the references provided in the response provides justification, this is not clear nor is it added to the manuscript.
References on the dose of MT (30 mg/kg), frequency (every two days) and route of administration (intraperitoneal injection) I have added in the Materials and Methods.
[15] Chen Z., Zhao C., Liu P., Huang H., Zhang S., Wang X. Anti-Apoptosis and Autophagy Effects of Melatonin Protect Rat Chondrocytes against Oxidative Stress via Regulation of AMPK/Foxo3 Pathways. Cartilage. 2021 ;13(2):1041S-1053S. doi: 10.1177/19476035211038748. 6
- The IHC methods provided read as a protocol and are not appropriate language for a manuscript. Further, while the authors acknowledged the importance of negative controls, they did not provide the requested information related to use of such, which is critical for evaluating the results in this section.
I have removed the IHC method from the article and provided it as an additional file.
In our experiments, the control group was defaulted to the sham group. I have modified the text to change the control group to the more accurate sham group. Usually, we use the sham group as a negative control group. In the sham group, only the skin was incised to open the joint cavity, but the cruciate lig-ament was not damaged. The sham group and the model group were intraperitoneally injected with a mixture of DMSO and saline. I have detailed it in Materials and Methods.
Due to the safety of DMSO and saline, we did not set it up as an independent control group
- Lines 182-184: While it is appreciated that the concentration of DMSO utilized for both the in vivo and in vitro work may be considered "safe" or "have no cytotoxic effects," this does not negate its importance as being more thoroughly considered as a negative control. It is also not clear if the model group received any IP injections?
The model group was given a mixture of normal saline and DMSO by intraperitoneal injection. I have detailed it in Materials and Methods.
Results:
- It is appreciated that improved images are provided for Figure 2. However, the language utilized to describe the findings for the model group do not match the Saf-O/Fast Green images. In addition, the areas evaluated in the 3 groups are at different regions within the joint. Measure bars should also be provided on the photomicrographs.
For the stained picture, I have changed to a more accurate description: The chondrocytes in the model group were distributed in clusters, and the red staining was significantly reduced. The number of vacuolated cells increased significantly, and the tide lines were distorted and irregular. Compared with the model group, the vacuolated cells in the MT group were significantly reduced. Although the chondrocytes were clustered, the matrix was uniformly stained red.
Menisci appeared in the control group due to visual field factors and the relatively thin cartilage in the control group. However, the images evaluated by safranin staining in the three groups were all tibial plateaus, which belonged to the same intra-articular region. Also, I re-added the measure bars to the microscope picture.
- It still remains a concern that the absence of specific control groups limits the interpretation of the in vitro findings. At a minimum, this would mean that the language surrounding interpretations should be less definitive and this limitation should be provided in the Discussion.
Due to the unclear expression of the control group and the sham group in the early stage, it caused difficulties in explaining the experimental results. Today, I express the control group exactly as the sham group. In the sham group, the joint cavity was opened, and the intraperitoneal injection of DMSO and saline was performed, excluding the influence of surgery and solvent on the experimental results. The sham group can be used as a control group to explain the experimental results.
Round 3
Reviewer 1 Report
Thank you for addressing some of the comments that were provided in the previous review. Remaining concerns (the majority of which have been repeated throughout previous reviews) can be found, below.
TItle: An update on the title is appreciated. However, "...in rats with experimentally induced osteoarthritis..." would be more appropriate.
Introduction:
Lines 50-52: “Animal models such as rats and pigs and in vitro experiments have confirmed that MT plays an important role in various ways such as intra-articular injection and intraperitoneal injection." Respectively, this change does not achieve the end intended by the authors.
Materials and Methods:
- Respectfully, the authors have still not addressed the specifics and/or use of negative controls in their IHC experiments. Further, as this reviewer does not see the location with the supplemental information pertinent to this section, this reviewer now must return to the original comment that, "Description of the IHC processes do not permit a reader to recapitulate the experiments. Further, there is no mention of a negative control, which is needed to appropriately quantitate immunostaining and account for background staining."
Results:
Lines 253-255: Language here has not been updated to match the new group descriptions. In this same section, DAB staining is typically considered a shade of brown, not yellow. Also, based on the representative images, "chondrons" may be a better description than "clusters" and "hypertrophied" may be better suited than "vacuolated."
Discussion:
Lines 407-408: "...the damage to the articular surface of the MT group was lighter" should be replaced by more appropriate terminology.
Line 435: Use of "...further verified it" should be replaced by, "To explore the mechanism of MT in OA, we further pursued such by in vitro experiments." This is suggested as monolayer culture is not necessarily replicative of in vivo conditions.
Lines 506-507: "MT can repair chondrocytes by activating the TGF-β1/Smad2 pathway to promote the expression of repair components, such as type II collagen." Use of "repair" in the first aspect of this sentence in reference to chondrocytes is a bit generic and should be replaced by something more specific and reflective of the results.
This reviewer would like to reiterate a comment provided in the original review: "One topic that would benefit this section would be to address the influence of MT at the dose utilized may have on the health and behavior of these rats (both of which can influence a multifaceted disease such as OA). In the absence of knowing what the MT did to these animals on a systemic level, a direct conclusion (versus an indirect influence) between MT and the in vivo results may be misleading." While the authors provided a response directly to this reviewer, none of this information ended up being addressed as a limitation in the Discussion.
Author Response
Many thanks to the reviewers for their efforts on this article. I am very sorry for the trouble I caused you due to my misunderstanding. I have made further changes to the article as you suggested.
TItle: An update on the title is appreciated. However, "...in rats with experimentally induced osteoarthritis..." would be more appropriate.
I have accepted your suggestion to change the title.
Introduction:
Lines 50-52: “Animal models such as rats and pigs and in vitro experiments have confirmed that MT plays an important role in various ways such as intra-articular injection and intraperitoneal injection." Respectively, this change does not achieve the end intended by the authors.
I have changed it to “In Aanimal models such as rats and pigs and in vitro experiments, efficacy of melatonin has been demonstrated following both intra-articular and intra-peritoneal injec-tions”
Materials and Methods:
- Respectfully, the authors have still not addressed the specifics and/or use of negative controls in their IHC experiments. Further, as this reviewer does not see the location with the supplemental information pertinent to this section, this reviewer now must return to the original comment that, "Description of the IHC processes do not permit a reader to recapitulate the experiments. Further, there is no mention of a negative control, which is needed to appropriately quantitate immunostaining and account for background staining."
I misunderstood the negative control you proposed in the previous period, and after careful study, I now realize my mistake and correct it. The focus of immunohistochemistry is that a negative control must be set for each antibody during the procedure. This is to rule out the presence of endogenous material or other specific coloration in a positive result. In this experiment, we used buffer instead of primary antibody as a negative control. I have provided the images of the negative controls as attachments. Also, In the material methods, I added a description of the negative control“We used buffer instead of primary antibody as a negative control”. In the results of immunohistochemistry I have stated “In immunohistochemistry, we used buffer instead of primary antibody as a negative control to exclude the effect of endogenous substances and specific staining on the experimental results”
Results:
Lines 253-255: Language here has not been updated to match the new group descriptions. In this same section, DAB staining is typically considered a shade of brown, not yellow. Also, based on the representative images, "chondrons" may be a better description than "clusters" and "hypertrophied" may be better suited than "vacuolated."
I have changed the language to the new group description. and I accept your suggestion to change to a more appropriate description.
Discussion:
Lines 407-408: "...the damage to the articular surface of the MT group was lighter" should be replaced by more appropriate terminology.
I have changed it to:The results of Safranin O fast green staining showed that compared with the model group, the morphology and extracellular matrix of chondrocytes in the MT group were relatively intact
Line 435: Use of "...further verified it" should be replaced by, "To explore the mechanism of MT in OA, we further pursued such by in vitro experiments." This is suggested as monolayer culture is not necessarily replicative of in vivo conditions.
I have accepted your suggestion and changed it.
Lines 506-507: "MT can repair chondrocytes by activating the TGF-β1/Smad2 pathway to promote the expression of repair components, such as type II collagen." Use of "repair" in the first aspect of this sentence in reference to chondrocytes is a bit generic and should be replaced by something more specific and reflective of the results.
I have changed it to:“MT protects chondrocytes by promoting the synthesis and secretion of type II colla-gen and reducing the degradation of ECM. This effect may be achieved by activating the TGF-β1/Smad2 pathway.”
This reviewer would like to reiterate a comment provided in the original review: "One topic that would benefit this section would be to address the influence of MT at the dose utilized may have on the health and behavior of these rats (both of which can influence a multifaceted disease such as OA). In the absence of knowing what the MT did to these animals on a systemic level, a direct conclusion (versus an indirect influence) between MT and the in vivo results may be misleading." While the authors provided a response directly to this reviewer, none of this information ended up being addressed as a limitation in the Discussion.
I added the effects of MT on animals and related literature to the discussion。:“Melatonin is an important hormone involved in the regulation of circadian rhythms, and several studies have shown that a decrease in systemic levels of melatonin can lead to circadian rhythm disorders[36,37]. In addition, circadian rhythm disorders are one of the causes of osteoarthritis, depression and many other diseases [38-40]. Here, we investigated the protective effect of melatonin on cartilage.”